# $\alpha$-TCVAE: On the relationship between Disentanglement and Diversity

**Cristian Meo**[*]        **Louis Mahon**                **Anirudh Goyal**              **Justin Dauwels**
TUDelft, NL.        University of Edinburgh, UK.    Google DeepMind, UK.    TUDelft, NL.

## Abstract

Understanding and developing optimal representations has long been founda-
tional in machine learning (ML). While disentangled representations have shown
promise in generative modeling and representation learning, their downstream
usefulness remains debated. Recent studies re-defined disentanglement through
a formal connection to symmetries, emphasizing the ability to reduce latent do-
mains (i.e., ML problem spaces) and consequently enhance data efficiency and
generative capabilities. However, from an information theory viewpoint, assign-
ing a complex attribute (i.e., features) to a specific latent variable may be infea-
sible, limiting the applicability of disentangled representations to simple datasets.
In this work, we introduce $\alpha$-TCVAE, a variational autoencoder optimized using
a novel total correlation (TC) lower bound that maximizes disentanglement and
latent variables informativeness. The proposed TC bound is grounded in informa-
tion theory constructs, generalizes the $\beta$-VAE lower bound, and can be reduced
to a convex combination of the known variational information bottleneck (VIB)
and conditional entropy bottleneck (CEB) terms. Moreover, we present quan-
titative analyses and correlation studies that support the idea that smaller latent
domains (i.e., disentangled representations) lead to better generative capabilities
and diversity. Additionally, we perform downstream task experiments from both
representation and RL domains to assess our questions from a broader ML per-
spective. Our results demonstrate that $\alpha$-TCVAE consistently learns more dis-
entangled representations than baselines and generates more diverse observations
without sacrificing visual fidelity. Notably, $\alpha$-TCVAE exhibits marked improve-
ments on MPI3D-Real, the most realistic disentangled dataset in our study, con-
firming its ability to represent complex datasets when maximizing the informative-
ness of individual variables. Finally, testing the proposed model off-the-shelf on
a state-of-the-art model-based RL agent, Director, significantly shows $\alpha$-TCVAE
downstream usefulness on the loconav Ant Maze task. Implementation available
at `https://github.com/Cmeo97/Alpha-TCVAE`

## 1 Introduction

The efficacy of machine learning (ML) algorithms is intrinsically tied to the quality of data repre-
sentation (Bengio et al., 2013). Such representations are useful not only for standard downstream
tasks such as supervised learning (Alemi et al., 2017) and reinforcement learning (RL) (Li, 2017),
but also for tasks such as transfer learning (Zhuang et al., 2020) and zero-shot learning (Sun et al.,
2021). Unsupervised representation learning aims to identify semantically meaningful represen-
tations of data without supervision, by capturing the generative factors of variations that describe
the structure of the data (Radford et al., 2016; Locatello et al., 2019b). According to Bengio et al.
(2013), disentanglement learning holds the key to understanding the world from observations, gen-
eralizing knowledge across different tasks and domains while learning and generating compositional
representations (Higgins et al., 2016; Kim & Mnih, 2018).

**Problem Formulation.** The goal of disentanglement learning is to identify a set of independent
generative factors $z$ that give rise to the observations $x$ via $p(x|z)$. However, from an information

---

[*]Work done while doing a research internship at Mila, Quebec AI Institute.

theory perspective, the amount of information retained by every latent variable may be insufficient to represent realistic generative factors (Kirsch et al., 2021; Do & Tran, 2020), limiting the applicability of disentangled representations to simple problems. What is more, Friedman & Dieng (2022) recently introduced the Vendi score, a new metric for gauging generative diversity, showing that entangled generative models, such as the Very Deep VAE (Child, 2020), consistently produce samples with less diversity compared to ground truth. This is indicative of their limited representational and generative prowess. In contrast, Higgins et al. (2019; 2022) re-defined disentangled representations through the lens of symmetries, linking disentanglement to computational problem spaces (e.g., disentangled representations inherently reduce the problem space (Arora & Barak, 2009)), suggesting that disentangled models should be able to explore and traverse the latent space more efficiently, leading to enhanced generative diversity.

**Previous Work.**   Most existing disentangled models optimize lower bounds that only impose an information bottleneck on the latent variables, and while this can result in factorized representations (Higgins et al., 2016), it does not directly optimize latent variable informativeness (Do & Tran, 2020). As a result, while several approaches have been proposed to learn disentangled representations by optimizing different bounds (Chen et al., 2018; Kim & Mnih, 2018), imposing sparsity priors (Mathieu et al., 2019), or isolating source of variance (Rolinek et al., 2019), none of the proposed models successfully learned disentangled representations of realistic datasets. Moreover, to the best of our knowledge, no systematic and quantitative analyses have been proposed to assess to what extent disentanglement and generative diversity (Friedman & Dieng, 2022) are correlated.

**Proposed method.**   In this work, we propose $\alpha$-TCVAE, a VAE optimized using a novel convex lower bound of the joint total correlation (TC) between the learned latent representation and the input data. The proposed bound, through a convex combination of the variational information bottleneck (VIB) Alemi et al. (2017) and the conditional entropy bottleneck (CEB) Fischer & Alemi (2020), maximizes the average latent variable informativeness, improving both representational and generative capabilities. The effectiveness of $\alpha$-TCVAE is especially prominent in the MPI3D-Real Dataset (Gondal et al., 2019), the most realistic dataset in our study that is compositionally built upon distinct generative factors. Figure 1 illustrates a comparison of the latent traversals between $\alpha$-TCVAE, Factor-VAE and $\beta$-VAE, showing that $\alpha$-TCVAE leads to the best visual fidelity and generative diversity (i.e., higher Vendi Score). Interestingly, the proposed TC bound is grounded in information theory constructs, generalizes the $\beta$-VAE (Higgins et al., 2016) lower bound, and can be reduced to a convex combination of the known variational information bottleneck (VIB) (Alemi et al., 2017) and conditional entropy bottleneck (CEB) (Fischer & Alemi, 2020) terms.

**Experimental Evaluation**   In order to determine the effectiveness of $\alpha$-TCVAE and the downstream usefulness of the learned representations, we measure the diversity and quality of generated images and disentanglement of its latent representations. Then, we perform a correlation study between the considered downstream scores across all models, analyzing how generative diversity and disentanglement are related across different datasets. This analysis substantiates our claim that disentanglement leads to improved diversity. Finally, we conduct experiments to assess the downstream usefulness of the proposed method from a broader ML perspective. Notably, the proposed method consistently outperforms the related baselines, showing a significant improvement in the RL Ant Maze task when applied off-the-shelf in Director, a hierarchical model-based RL agent (Hafner et al., 2022).

## 2   RELATED WORK

**Generative Modelling and Disentanglement**   Recently Locatello et al. (2019b) demonstrated that unsupervised disentangled representation learning is theoretically impossible, nonetheless disentangled VAEs, acting as both representational and generative models, Kingma & Welling (2013); Higgins et al. (2016); Chen et al. (2018); Kim & Mnih (2018) achieve practical results by leveraging implicit biases within the data and learning dynamics Burgess & Kim (2018); Higgins et al. (2019); Mathieu et al. (2019). On the generation side, they have been widely used to generate data such as images (Chen et al., 2019), text (Shi et al., 2019), speech (Sun et al., 2020; Li et al., 2023) and music (Wang et al., 2020). Various extensions to the base VAE model have been presented to improve generation quality in terms of visual fidelity (Peng et al., 2021; Vahdat & Kautz, 2020; Razavi et al.,

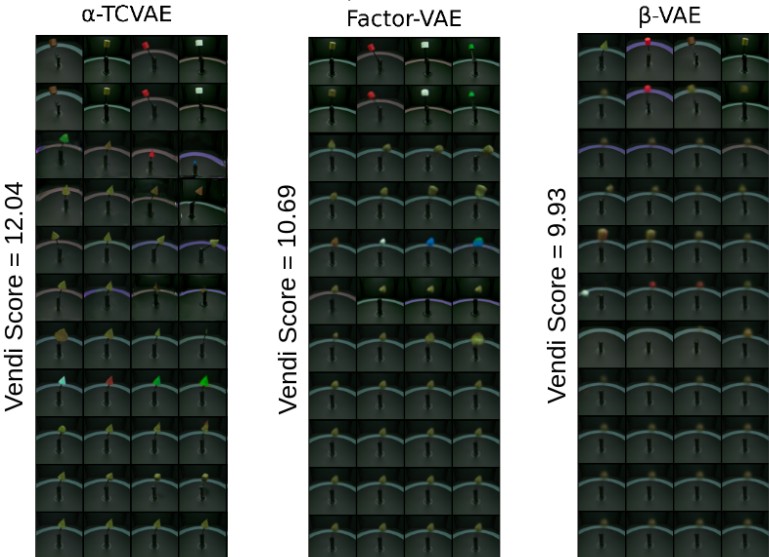

Figure 1: Ground truth (first row), reconstructions (second row) and latent traversals comparison of $\alpha$-TCVAE, Factor-VAE, and $\beta$-VAE on the MPI3D-Real Dataset. Notably, $\alpha$-TCVAE showcases superior visual fidelity and generative diversity, as indicated by a higher Vendi Score.

2019). On the representational side, aiming for explainable and factorized representations, Higgins et al. (2016) proposed $\beta$-VAE, which inspired a number of following disentangled VAE-based models, such as Factor-VAE (Kim & Mnih, 2018), $\beta$-TCVAE Chen et al. (2018), and $\beta-$Annealed VAE (Burgess et al., 2018). Both $\beta$-VAE and Factor-VAE aim to learn disentangled representations by imposing a bottleneck on the information flowing through the latent space. While $\beta$-VAE does this by introducing a $\beta$ hyperparameter that increases the strength of the information bottleneck, Factor-VAE introduces a TC regularization term. Chen et al. (2018) proposed $\beta$-TCVAE, which minimizes the total correlation of the latent variables using Monte-Carlo and importance sampling. Roth et al. (2023) proposed the Hausdorff Factorized Support (HFS) criterion, a relaxed disentanglement criterion that encourages only pairwise factorized support, rather than a factorial distribution, by minimizing a Hausdorff distance. This allows for arbitrary distributions of the factors over their support, including correlations between them. Our model, namely $\alpha$-TCVAE is optimized by a TC lower bound as well, however we do not make use of any trick or expensive sampling strategy. In contrast, we derive a TC lower bound that does not require any extra network or sampling strategy and is theoretically grounded in the Deep Information Bottleneck framework Alemi et al. (2017).

**Disentanglement and Deep Information Bottleneck** In the last few years, a link between the latent space capacity and disentanglement of the learned variables (Bengio et al., 2013; Shwartz-Ziv & Tishby, 2017; Goyal et al., 2021) has been identified, showing that decreasing the capacity of a network induces disentanglement on the learned representations. This relationship has been explained by the information bottleneck (IB) principle, introduced by Tishby et al. (2001) as a regularization method to obtain minimal sufficient encoding by constraining the amount of information captured by the latent variables from the observed variable. Variational IB (VIB) (Alemi et al., 2017) has extended the IB framework by applying it to neural networks, which results in a simple yet effective method to learn representations that generalize well and are robust against adversarial attacks. Furthermore, (Alemi et al., 2017; Kirsch et al., 2021) outlined the relationship between VIB, VAE (Kingma & Welling, 2013) and $\beta$-VAE (Higgins et al., 2016), providing an information theoretical interpretation of the Kullback-Leibler (KL) divergence term used in these models as a regularizer. Despite the advantages introduced by the VIB framework, imposing independence between every latent variable may be too strong an assumption (Roth et al., 2023). For this reason, Fischer & Alemi (2020) introduced the conditional entropy bottleneck (CEB), which assumes conditional independence between the learned latent variables, providing the ability to learn more expressive and robust representations (Kirsch et al., 2021). Recently, a generalization of the mutual Information (MI),

namely total correlation (TC), has been used to learn disentangled representations as well (Kim & Mnih, 2018). Following Hwang et al. (2021), who propose a similar TC bound for a multi-view setting, we derive a novel TC lower bound for the unsupervised representational learning setting. As a result, the proposed bound is able to learn expressive and disentangled representations.

**Disentanglement and Diversity**   The ideal generative model learns a distribution that well explains the observed data, which can then be used to draw a diverse set of samples. Diversity is thus an important desirable property of generative models (Friedman & Dieng, 2022). We desire the ability to produce samples that are different from each other and from the samples we already have at train time, while still coming from the same underlying distribution. The benefits of diversity have been advocated in a number of different contexts, such as image synthesis (Mao et al., 2019), molecular design (Lin et al., 2021; Wu et al., 2021), natural language text (McCoy et al., 2023), and drug discovery (Kim & Mnih, 2018). Motivated by the benefits of generative diversity, several VAE-based models have aimed to show increased diversity in their generated samples (Razavi et al., 2019). Some works have also noted improvements in diversity due to disentanglement. Lee et al. (2018) adversarially disentangle style from content and show enhanced diversity of image-to-image translations. Kazemi et al. (2019) also perform style-content disentanglement, this time in the context of text generation, and again observe an increase in diversity. Li et al. (2020) shows that disentangling pose, shape, and texture leads to greater diversity in generated images. Collectively, these studies emphasize that diversity is often a valuable indicator of effectiveness in various applications, and suggest that diversity and disentanglement are intertwined aspects of generative models. Yet, to the best of our knowledge, no quantitative analyses that support this claim have been presented. In this work, we present a correlation study, showing how downstream metrics of disentanglement (e.g., DCI (Eastwood & Williams, 2018)) and diversity (e.g., Vendi Score (Friedman & Dieng, 2022)) are correlated across several models and datasets.

## 3   $\alpha$-TCVAE FRAMEWORK DERIVATION

**Motivation.**   In contrast to most existing methods, which only impose an information bottleneck to learn disentangled representations, we seek to maximize the informativeness of individual latent variables as well. The total joint correlation (TC) can be explicitly expressed in terms of mutual information between the observed data and the latent generative factors, as shown in equation 4, allowing us to link disentanglement to latent variables informativeness. As a result, leveraging the TC formulation, we can derive a lower bound that not only promotes disentanglement but also maximizes the information retained by individual latent variables.

**Derivation.**   In this section, we formally derive the novel TC bound. Let $\mathcal{D} = \{\boldsymbol{X}, \boldsymbol{V}\}$ be the ground-truth set that consists of images $\boldsymbol{x} \in \mathbb{R}^{N \times N}$, and a set of conditionally independent ground-truth data generative factors $\boldsymbol{v} \in \mathbb{R}^{M}$, where $\log p(\boldsymbol{v}|\boldsymbol{x}) = \sum_k \log p(v_k|\boldsymbol{x})$. The goal is to develop an unsupervised deep generative model that can learn the joint distribution of the data $\boldsymbol{x}$, while uncovering a set of generative latent factors $\boldsymbol{z} \in \mathbb{R}^{K}$, $K \geq M$, such that $\boldsymbol{z}$ can fully describe the data structure of $\boldsymbol{x}$ and generate data samples that follow the underlying ground-truth generative factors $\boldsymbol{v}$. Since directly optimizing the joint TC is intractable, we are going to maximize a lower bound of the joint total correlation $TC(\boldsymbol{z}, \boldsymbol{x})$ between the learned latent representations $\boldsymbol{z}$ and the input data $\boldsymbol{x}$, following the approach proposed by Hwang et al. (2021). The total correlation is defined as the KL divergence between the joint distribution and the factored marginals. In our case:

$$TC_\theta(\boldsymbol{z}) \triangleq D_{KL}\left[\int q_\theta(\boldsymbol{z}|\boldsymbol{x})p_D(\boldsymbol{x})d\boldsymbol{x} \| \prod_{k=1}^{K} q_\theta(\boldsymbol{z}_k)\right], \tag{1}$$

where the joint distribution is $q_\theta(\boldsymbol{z}) = \int q_\theta(\boldsymbol{z}|\boldsymbol{x})p_D(\boldsymbol{x})d\boldsymbol{x}$, $p_D(\boldsymbol{x})$ is the data distribution, $q_\theta(\boldsymbol{z}_k) = \int q_\theta(\boldsymbol{z}|\boldsymbol{x})d\boldsymbol{z}_{\neq k}$ and $\boldsymbol{z}_{\neq k}$ indicates that the k-th component of $\boldsymbol{z}$ is not considered. Since we aim to find the encoder $q_\theta(\boldsymbol{z}|\boldsymbol{x})$ that disentangles the learned representations $\boldsymbol{z}$, we can formulate the following objective:

$$TC_\theta(\boldsymbol{z}, \boldsymbol{x}) \triangleq TC_\theta(\boldsymbol{z}) - TC_\theta(\boldsymbol{z}|\boldsymbol{x}), \tag{2}$$

where the conditional TC$(\boldsymbol{z}|\boldsymbol{x})$ can be expressed as:

$$TC_\theta(\boldsymbol{z}|\boldsymbol{x}) \triangleq \mathbb{E}_{q_\theta(\boldsymbol{z})}\left[D_{KL}\left[q_\theta(\boldsymbol{z}|\boldsymbol{x}) \| \prod_{k=1}^{K} q_\theta(\boldsymbol{z}_k|\boldsymbol{x})\right]\right], \tag{3}$$

which is the expected KL divergence of the joint conditional from the factored conditionals. Intuitively, we can see that minimizing $TC_\theta(\boldsymbol{z}|\boldsymbol{x})$, $TC_\theta(\boldsymbol{z}, \boldsymbol{x})$ will be maximized, enhancing the disentanglement of the learned representation $\boldsymbol{z}$. Moreover, decomposing equation 2 we can express the TC in terms of MI (Gao et al., 2019):

$$TC_\theta(\boldsymbol{z}, \boldsymbol{x}) = \sum_{k=1}^{K} I_\theta(\boldsymbol{z}_k, \boldsymbol{x}) - I_\theta(\boldsymbol{z}, \boldsymbol{x}), \tag{4}$$

where $I_\theta(\boldsymbol{z}, \boldsymbol{x})$ is the mutual information between $\boldsymbol{z}$ and $\boldsymbol{x}$ and is known as the VIB term Alemi et al. (2017). Additionally, we can also express it in terms of Conditional MI:

$$TC_\theta(\boldsymbol{z}, \boldsymbol{x}) = \frac{1}{K} \sum_{k=1}^{K} \left[ (K-1) I_\theta(\boldsymbol{z}_k, \boldsymbol{x}) - I_\theta(\boldsymbol{z}_{\neq k}, \boldsymbol{x}|\boldsymbol{z}_k) \right], \tag{5}$$

where $I_\theta(\boldsymbol{z}_{\neq k}, \boldsymbol{x}|\boldsymbol{z}_k)$ is known as the CEB term (Fischer & Alemi, 2020). Equation 4 and equation 5 illustrate the link of the designed objective to both VIB and CEB frameworks. A complete derivation of them can be found in Appendices A.1 and A.2, respectively. While the VIB term promotes compression of the latent representation, the CEB term promotes balance between the information contained in each latent dimension. Since we want to promote both disentanglement and individual variable informativeness of the learned latent representation we propose a lower bound that convexly combines the found VIB and CEB terms. We define the bound as follows:

$$TC(\boldsymbol{z}, \boldsymbol{x}) \geq \mathbb{E}_{q_\theta(\boldsymbol{z}|\boldsymbol{x})} \left[ \log p_\phi(\boldsymbol{x}|\boldsymbol{z}) \right] - \underbrace{\frac{K\alpha}{K-\alpha} D_{KL}(q_\theta(\boldsymbol{z}|\boldsymbol{x}) \| r_p(\boldsymbol{z}|\boldsymbol{x}))}_{\text{CEB}} - \underbrace{\frac{(1-\alpha)}{(1-\frac{\alpha}{K})} D_{KL}(q_\theta(\boldsymbol{z}|\boldsymbol{x}) \| r(\boldsymbol{z}))}_{\text{VIB}}, \tag{6}$$

where $\alpha$ is a hyperparameter that trades off VIB and CEB terms. Following Hwang et al. (2021), we define $r_p(\boldsymbol{z}|\boldsymbol{x}) = N(\boldsymbol{\mu}_p, \boldsymbol{\sigma}_p \boldsymbol{I})$ and $r(\boldsymbol{z}) = N(\boldsymbol{0}, \boldsymbol{I})$, respectively, where $\boldsymbol{\sigma}_p \triangleq \left( \sum_{k=1}^{K} \frac{1}{\boldsymbol{\sigma}_k^2} \right)^{-1}$ and $\boldsymbol{\mu}_p \triangleq \boldsymbol{\sigma}_p \cdot \sum_{k=1}^{K} \frac{\boldsymbol{\mu}_k}{\boldsymbol{\sigma}_k^2}$ while $\boldsymbol{\mu}_k$ and $\boldsymbol{\sigma}_k$ are the mean and standard deviation used to compute $\boldsymbol{z}_k$ using the reparametrization trick as in Kingma & Welling (2013). A full derivation of the bound defined in equation 6 can be found in Appendix A.

**Practical Implications.** Disentangled models with $M$ generative factors and $K$ latent dimensions usually have $(K - M)$ noisy latent dimensions Do & Tran (2020), but our CEB term induces an inductive bias on the information flowing through every individual latent variable, pushing otherwise noisy dimensions to be informative. The derived TC lower bound generalizes the structure of the widely used $\beta$-VAE (Higgins et al., 2016) bound. Indeed, for $\alpha = 0$, the TC bound reduces to $\beta$-VAE one. A comparison of $\alpha$-TCVAE, $\beta$-VAE, $\beta$-TCVAE, HFS and Factor-VAE lower bounds can be found in Tab. A.3.

## 4 EXPERIMENTS

In this section, we design empirical experiments to understand the performance of $\alpha$-TCVAE and its potential limitations by exploring the following questions: (1) Does maximising the informativeness of latent variables consistently lead to an increase in representational power and generative diversity? (2) Do disentangled representations inherently present higher diversity than entangled ones? (3) How are they correlated with other downstream metrics (i.e., FID (Heusel et al., 2017) and unfairness (Locatello et al., 2019a))? (4) To what extent does maximising the latent variables' informativeness in disentangled representations improve their downstream usefulness?

**Experimental Setup**. In order to assess the performance of both proposed and baseline models, we validate the considered models on the following datasets. **Teapots** (Moreno et al., 2016) contains $200,000$ images of teapots with features: azimuth and elevation, and object colour. **3DShapes** (Burgess & Kim, 2018) contains $480,000$ images, with features: object shape and colour, floor colour, wall colour, and horizontal orientation. **MPI3D-Real** (Gondal et al., 2019) contains $103,680$ images of objects at the end of a robot arm, with features: object colour, size, shape, camera height, azimuth, and robot arm altitude. **Cars3D** (Reed et al., 2015) contains $16,185$ images with features: car-type, elevation, and azimuth. **CelebA** (Liu et al., 2015) contains over $200,000$ images of faces

under a broad range of poses, facial expressions, and lighting conditions, totalling 40 different factors. All datasets under consideration consist of RGB images with dimensions $64 \times 64$. Among them, CelebA stands out as the most realistic and complex dataset. On the other hand, MPI3D-Real is considered the most realistic among factorized datasets, which we define as those compositionally generated using independent factors. To assess the generated images, we use the FID score (Heusel et al., 2017) to measure the distance between the distributions of generated and real images, and the Vendi score (Friedman & Dieng, 2022) to measure the diversity of sampled images. Both Vendi and FID use the Inception Network (Szegedy et al., 2017) to extract image features and compute the related similarity metrics. Since **DCI** (Eastwood & Williams, 2018) scores can produce unreliable results in certain cases, (Mahon et al., 2023; Cao et al., 2022; Do & Tran, 2020), we measure disentanglement using also single neuron classification SNC (Mahon et al., 2023). Further details on used datasets and metrics are given in Appendix C.

**Baseline Methods**. We compare $\alpha$-TCVAE to four other VAE models: $\beta$-VAE (Higgins et al., 2016), $\beta$-TCVAE (Chen et al., 2018), $\beta$-VAE+HFS (Roth et al., 2023) and FactorVAE (Kim & Mnih, 2018), all of which are described in Section 2, as well as a vanilla VAE (Kingma & Welling, 2013). To assess diversity and visual fidelity beyond VAE-based models, we also compare to a generative adversarial network model, StyleGAN (Karras et al., 2019).

**Generation Faithfulness and Diversity Analyses**. We present image generation results from our model alongside baseline models, evaluating performance on the FID and Vendi metrics across datasets. For image generation using VAE-models, we adopt two strategies: (1) Sampling a noise vector from a multivariate standard normal and decoding it. (2) Encoding an actual image, then selecting a latent dimension. The value of this chosen dimension is adjusted by shifts of $+/-$ 1, 2, 4, 6, 8, or 10 standard deviations. Subsequently, we decode the adjusted representation. In Figures 2 and 3 the two sampling strategies are labeled as 'Sampled from Noise' and 'Sampled from Traversals' respectively. Figures 2 and 3 show that $\alpha$-TCVAE consistently generates more diverse (higher Vendi) and more faithful (lower FID) images than baseline VAE models.

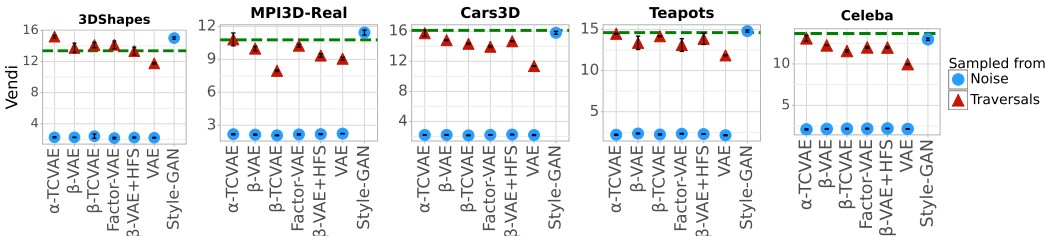

Figure 2: Diversity of generated images, as measured by Vendi score. Two different sampling strategies are considered: sampled from white noise and from traversals. The diversity of the images of our model, $\alpha$-TCVAE, is consistently higher than baseline VAE models, and on par with StyleGAN. The green dashed line represents ground truth dataset diversity. Traversals produce significantly more diverse images than samples.

The Vendi score of $\alpha$-TCVAE is comparable to that of Style-GAN, and its FID score is only slightly worse. Moreover, Style-GAN takes 15x the training time ($\sim$ 2hrs vs. $>$ 30hrs on a single Nvidia Titan XP) and learns only a generative model, whereas VAEs learn both a generative model and a representational model. Noticeably, all VAE-based models perform poorly in terms of both diversity and reconstruction quality when sampling from white noise, highlighting the benefit of a structured sampling strategy when using VAE-based models for generative tasks. Another finding is that traversal-generated images are superior to those obtained from the prior, i.e. sampling from a standard normal and decoding. This is in keeping with prior work showing that drawing latent samples from a distribution other than the standard normal, e.g. a GMM, often leads to higher quality generated images Chadebec et al. (2022), and it supports the claim that disentangled models allow more systematic exploration of the latent space leading to more diverse images. This claim is also supported by noting that all disentangled VAEs give higher diversity than the vanilla VAE.

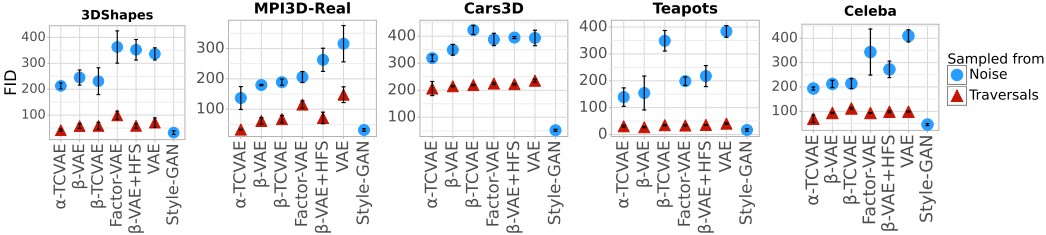

Figure 3: Faithfulness of generated images to the data distribution, as measured by FID score. Two different sampling strategies are considered: sampled from white noise and from traversals. The scores for the images of our model, $\alpha$-TCVAE, are consistently better than baseline VAE models (lower FID is better), and only slightly worse than StyleGAN. Traversals produce significantly more faithful images than samples.

**Disentanglement Analyses and Downstream Metrics Correlation Study**    In this section we examine the disentanglement capabilities of $\alpha$-TCVAE and the related VAE baselines, and how it relates, statistically, with the diversity and quality of generated images, as measured in Section 4. Figures 4, 5 and 6 show that $\alpha$-TCVAE consistently achieves comparable or better DCI, SNC and unfairness scores. The improvement of $\alpha$-TCVAE over the baselines is most significant on the most realistic factorized dataset, namely MPI3D-Real. Interestingly, while there is a significant gap between the DCI scores of disentangled and entangled models across every factorized dataset, SNC shows that in terms of single neuron factorization, for both Cars3D and MPI3D-Real, $\alpha$-TCVAE is the only model that significantly improves over the entangled VAE. This is perhaps due to the tendency of DCI to sometimes overestimate disentanglement Mahon et al. (2023); Cao et al. (2022).

Furthermore, as illustrated in Figure 4, no model has been successful in learning disentangled representations from the CelebA dataset. To meaningfully encode CelebA images, we used high-dimensional latent representations (e.g., 48 dimensions). However, as highlighted by Do & Tran (2020), disentangling and measuring disentanglement in high-dimensional representations are notoriously challenging tasks. Indeed, while DCI and unfairness present unrealistic results, SNC gave all models a score of zero, and so we do not display the figures here. Figure 10 illustrates a significant correlation between the Vendi, unfairness, and DCI metrics. There is a compelling correlation between Vendi and DCI scores, underscoring that diversity and disentanglement are statistically related. This resonates with the understanding that disentangled latent spaces naturally exhibit superior generative diversity (Higgins et al., 2019). Additionally, Vendi and DCI both exhibit a negative correlation with unfairness. This observation is consistent with Locatello et al. (2019a)'s findings, implying that the fairness of downstream prediction tasks is deeply associated with the diversity and disentanglement of the representations being learned. Further correlations results are given in Appendix D, along with examples of latent traversals.

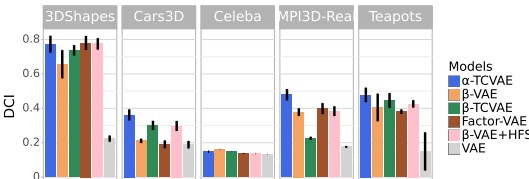

Figure 4: Comparison of DCI scores of our model with those of baseline models.

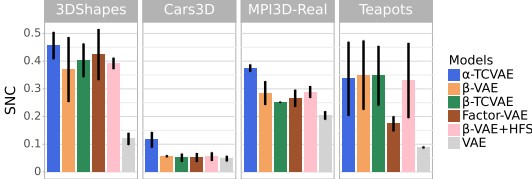

Figure 5: Comparison of SNC scores of our model with those of baseline models.

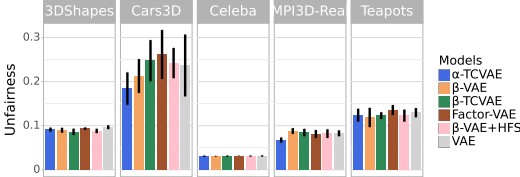

Figure 6: Comparison of unfairness scores of our model with those of baseline models.

**Attribute Classification Task**    In this experiment, we train a multilayer perceptron (MLP)

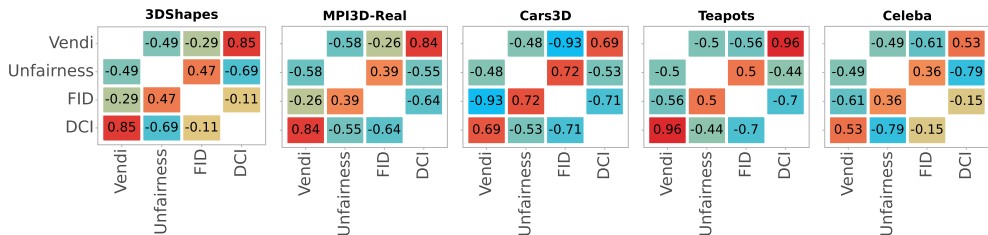

Figure 7: Correlations between diversity (Vendi score), generation faithfulness (FID score), unfairness and DCI. Correlations are computed using the results from all models across 5 different seeds.

to classify sample attributes using the models' encoded latent representations. Figure 8 reveals that $\alpha$-TCVAE either matches or surpasses the baseline models in terms of attribute classification accuracy. The improvement is minor on 3DShapes and Teapots, but more significant on Cars3D and MIP3D-Real. Interestingly, the only dataset where all VAEs exhibit commendable performance is CelebA, where high-dimensional representations are used. This suggests that, for this particular downstream task, the dimensionality of the representation may be the main constraining factor. In fact, this downstream task inherently favours high-dimensional attributes, considering that a MLP is employed for the attribute classification.

**Loconav Ant Maze Reinforcement Learning Task**. In this experiment, a model-based RL agent has to learn its proprioceptive dynamical system while escaping from a maze. Recently, Hafner et al. (2022) introduced Director, a hierarchical model-based RL agent. Director employs a hierarchical strategy with a Goal VAE that learns and generates sub-goals, simplifying the planning task. The first hierarchy level represents the agent's internal states, while in the second one, the Goal VAE encodes the agent's state and infers sub-goals. As a result, the Goal VAE generates sub-goals to guide the agent through the environment. Given the enhanced generative diversity of $\alpha$-TCVAE, we postulated that integrating our proposed TC bound could improve Director's exploration. In

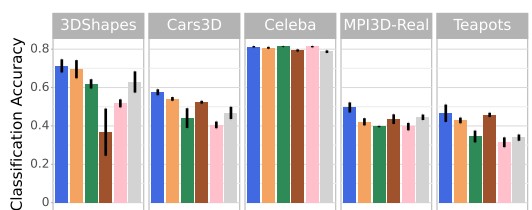

Figure 8: Comparison of $\alpha$-TCVAE and baseline models on the Downstream Attribute Classification Task. Our proposed model either matches or surpasses the baseline models in terms of attribute classification accuracy

this experiment, we replaced the beta-VAE objective, used to train Director's Goal VAE, with our TC-bound, expecting a richer diversity in sub-goals, thus expediting environment exploration and enhancing overall learning behaviour. Figure 9 compares the performance of Director and Alpha-

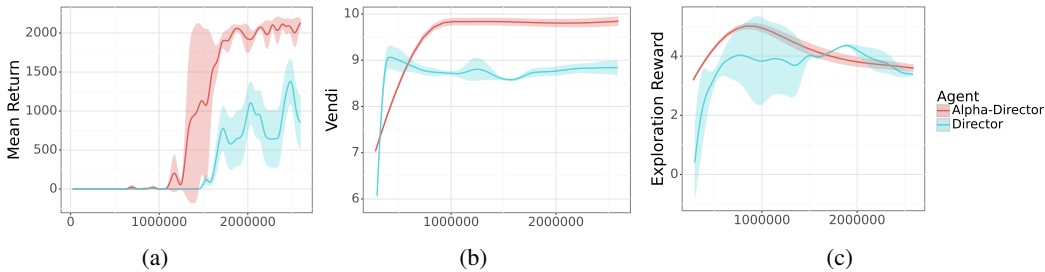

Figure 9: Performance of Director, a model-based hierarchical RL agent, and Alpha-Director on the Antmaze task. While director samples sub-goals using the original $\beta$-VAE, Alpha-Director samples sub-goals using the proposed $\alpha$-TCVAE. Sampling using $\alpha$-TCVAE gives more diverse goals (b), better exploration (c) and significantly higher mean return (a).

Director, which replaces $\beta$-VAE objective with the proposed TC-bound instead, the results are averaged across three seeds. Figure 9-(a) presents the mean return, which scores the performances of the agent on the given task (i.e., finding the exit of the maze while learning proprioceptive dynamics), showing that Alpha-Director significantly outperforms Director, learning faster and to a higher final high mean reward. Figure 9-(b) illustrates the Vendi score of sampled goals across batch and sequence length, showing that $\alpha$-TCVAE generates sub-goals with a higher diversity score. As a result, Alpha-Director has a better exploration, as shown in Figure 9-(c), leading to faster learning. Collectively, these findings highlight that $\alpha$-TCVAE enables the agent to sample a broader range of sub-goals, fostering efficient exploration and ultimately enhancing task performance.

## 5 Discussion and Future Work

Through comprehensive quantitative analyses, we answer the defined research questions while delineating the advantages and limitations of the proposed model relative to the evaluated baselines. Our findings resonate with the hypothesis posited by Higgins et al. (2019), emphasizing a strong correlation between disentanglement and generative diversity. Notably, disentangled representations consistently showcase enhanced visual fidelity and diversity compared to the entangled ones. This correlation persists across all datasets rendered using disentangled representations. Intriguingly, traversal analyses of $\alpha$-TCVAE, illustrated in Figures 1 and 16 in Appendix C, reveal that it is able to discover novel generative factors, such as object positioning and vertical perspectives, which are absent from the training dataset. We hypothesize that the CEB term is responsible for this phenomenon. Most existing models optimize only the information bottleneck, and while this can result in factorized representations, it does not directly optimize latent variable informativeness. Our proposed bound also includes a CEB term, and so maximizes the average informativeness as well, which may push otherwise noisy variables to learn new generative factors. Future research will delve deeper into comprehending this phenomenon and exploring its potential applications.

In accordance with the literature, the main limitation of $\alpha$-TCVAE is that, akin to other disentangled VAEs, it is difficult to scale efficiently. This scaling challenge permeates the entire disentanglement paradigm. In high-dimensional spaces, not only do disentangled VAE-based models struggle to produce disentangled representations, but also the metrics used to measure disentanglement tend not to be useful. (e.g., DCI and SNC(Eastwood & Williams, 2018; Mahon et al., 2023)). On the other hand, disentangled representations have a number of desirable properties, as already showcased in the literature (Higgins et al., 2022). In particular, their impact is undeniable in the Ant Maze RL experiment from Figure 9. Reinforcing this observation, our correlation study underscores the relationship between disentanglement and diversity, leading to the following question: can we leverage diversity as a surrogate for measuring disentanglement in complex and high-dimensional scenarios? We leave the answer to this question as a future work.

## 6 Conclusion

We introduce $\alpha$-TCVAE, a VAE optimized through a convex lower bound on the joint total correlation (TC) between the latent representation and the input data. This proposed bound naturally reduces to a convex combination of the known variational information bottleneck (VIB) (Alemi et al., 2017) and the conditional entropy bottleneck (CEB) (Fischer & Alemi, 2020). Moreover, it generalizes the widely adopted $\beta$-VAE bound. By maximizing disentanglement and average informativeness of the latent variables, our approach enhances both representational and generative capabilities. A comprehensive quantitative evaluation indicates that $\alpha$-TCVAE consistently produces superior representations. This is evident from its performance across key downstream metrics: disentanglement (i.e., DCI and SNC), generative diversity (i.e., Vendi score), visual fidelity (i.e., FID), and its demonstrated downstream usefulness. In particular, our $\alpha$-TCVAE showcases significant improvements on the MPI3D-Real dataset, the most realistic factorized dataset in our evaluation, and in a downstream reinforcement learning task. This highlights the strength of maximizing the average informativeness of latent variables, offering a pathway to address the inherent challenges of disentangled VAE-based models.

## 7 ETHIC STATEMENT AND REPRODUCIBILITY

To the best of the authors' knowledge, this study does not involve any ethical issues. The authors aim to maximize the reproducibility of the study. The codes of this project will be released in the camera-ready version. In the methods section, notions align with existing literature.

## 8 ACKNOWLEDGEMENTS

We thank Prof. Yoshua Bengio for the useful feedback provided along the project. We thank Mila - Quebec AI institute, TUDelft, and Compute Canada for providing all the resources to make the project possible.

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

# A    TOTAL CORRELATION LOWER BOUND DERIVATION

In this section we are going to derive the TC lower bound defined in equation 6. Since it is defined as a convex combination of marginal log-likelihood, VIB, and CEB terms, we are going to split the derivation into two subsections. First, we will derive a first TC bound that introduces the VIB term. Then, we will derive another TC bound, which explicitly shows the CEB term. Finally, we will define the TC bound shown in equation 6 as a convex combination of the two bounds.

## A.1    TC BOUND AND THE VARIATIONAL INFORMATION BOTTLENECK

Unfortunately, direct optimization of mutual information terms is intractable Alemi et al. (2017). Therefore, we first need to find a lower bound of equation 4. Following the approach used in Hwang et al. (2021), we can expand it as:

$$
\begin{aligned}
TC_\theta(\boldsymbol{z}, \boldsymbol{x}) &= \sum_{k=1}^{K} I_\theta(\boldsymbol{z}_k, \boldsymbol{x}) - I_\theta(\boldsymbol{z}, \boldsymbol{x}), \\
&= \sum_{k=1}^{K} \left[ \mathbb{E}_{q_\theta(\boldsymbol{x}, \boldsymbol{z}_k)} \left[ \log \frac{q_\theta(\boldsymbol{x}|\boldsymbol{z}_k)}{p_D(\boldsymbol{x})} \right] \right] - \mathbb{E}_{q_\theta(\boldsymbol{x}, \boldsymbol{z})} \left[ \log \frac{q_\theta(\boldsymbol{x}|\boldsymbol{z})}{p_D(\boldsymbol{x})} \right], \\
&= \sum_{k=1}^{K} \left[ \mathbb{E}_{q_\theta(\boldsymbol{x}, \boldsymbol{z}_k)} \left[ \log \frac{q_\theta(\boldsymbol{x}|\boldsymbol{z}_k)}{p_D(\boldsymbol{x})} \frac{p_\phi(\boldsymbol{x}|\boldsymbol{z}_k)}{p_\phi(\boldsymbol{x}|\boldsymbol{z}_k)} \right] \right] - \mathbb{E}_{q_\theta(\boldsymbol{z}, \boldsymbol{x})} \left[ \log \frac{q_\theta(\boldsymbol{z}|\boldsymbol{x})}{q_\theta(\boldsymbol{z})} \frac{r(\boldsymbol{z})}{r(\boldsymbol{z})} \right].
\end{aligned}
\tag{7}
$$

Let's expand these two terms:

$$
\begin{aligned}
\mathbb{E}_{q_\theta(\boldsymbol{x}, \boldsymbol{z}_k)} \left[ \log \frac{q_\theta(\boldsymbol{x}|\boldsymbol{z}_k)}{p_D(\boldsymbol{x})} \frac{p_\phi(\boldsymbol{x}|\boldsymbol{z}_k)}{p_\phi(\boldsymbol{x}|\boldsymbol{z}_k)} \right] &= \int \int q_\theta(\boldsymbol{z}_k, \boldsymbol{x}) \log \frac{q_\theta(\boldsymbol{x}|\boldsymbol{z}_k)}{p_D(\boldsymbol{x})} \frac{p_\phi(\boldsymbol{x}|\boldsymbol{z}_k)}{p_\phi(\boldsymbol{x}|\boldsymbol{z}_k)} d\boldsymbol{z}_k d\boldsymbol{x}, \\
&= \int \int q_\theta(\boldsymbol{z}_k|\boldsymbol{x}) p_D(\boldsymbol{x}) \left( \log \left( \frac{q_\theta(\boldsymbol{x}|\boldsymbol{z}_k)}{p_\phi(\boldsymbol{x}|\boldsymbol{z}_k)} \right) + \log p_\phi(\boldsymbol{x}|\boldsymbol{z}_k) - \log p_D(\boldsymbol{x}) \right) d\boldsymbol{z}_k d\boldsymbol{x}, \\
&= H(\boldsymbol{x}) + \mathbb{E}_{q_\theta(\boldsymbol{z}_k)} [D_{KL}(q_\theta(\boldsymbol{x}|\boldsymbol{z}_k)\|p_\phi(\boldsymbol{x}|\boldsymbol{z}_k))] + \mathbb{E}_{q_\theta(\boldsymbol{z}_k, \boldsymbol{x})} [\log p_\phi(\boldsymbol{x}|\boldsymbol{z}_k)].
\end{aligned}
\tag{8}
$$

$$
\begin{aligned}
\mathbb{E}_{q_\theta(\boldsymbol{z}, \boldsymbol{x})} &\left[ \log \frac{q_\theta(\boldsymbol{z}|\boldsymbol{x})}{q_\theta(\boldsymbol{z})} \frac{r(\boldsymbol{z})}{r(\boldsymbol{z})} \right], \\
&= \int q_\theta(\boldsymbol{x}, \boldsymbol{z}) \log \left( \frac{q_\theta(\boldsymbol{z}|\boldsymbol{x})}{q_\theta(\boldsymbol{z})} \frac{r(\boldsymbol{z})}{r(\boldsymbol{z})} \right) d\boldsymbol{z} d\boldsymbol{x}, \\
&= \int q_\theta(\boldsymbol{z}|\boldsymbol{x}) p_D(\boldsymbol{x}) \left( \left( \log \frac{q_\theta(\boldsymbol{z}|\boldsymbol{x})}{r(\boldsymbol{z})} \right) + \log \left( \frac{r(\boldsymbol{z})}{q_\theta(\boldsymbol{z})} \right) \right) d\boldsymbol{z} d\boldsymbol{x}, \\
&= \mathbb{E}_{p_D(\boldsymbol{x})} [D_{KL}(q_\theta(\boldsymbol{z}|\boldsymbol{x})\|r(\boldsymbol{z}))] - \mathbb{E}_{q_\theta(\boldsymbol{x}|\boldsymbol{z})} [D_{KL}(q_\theta(\boldsymbol{z})\|r(\boldsymbol{z}))].
\end{aligned}
\tag{9}
$$

As a result, we can write:

$$TC_\theta(\boldsymbol{z}, \boldsymbol{x}) = \sum_{k=1}^{K} \left[ H(\boldsymbol{x}) + \mathbb{E}_{q_\theta(\boldsymbol{z}_k)}[D_{KL}(q_\theta(\boldsymbol{x}|\boldsymbol{z}_k)\|p_\phi(\boldsymbol{x}|\boldsymbol{z}_k))] + \mathbb{E}_{q_\theta(\boldsymbol{z}_k, \boldsymbol{x})}[\log p_\phi(\boldsymbol{x}|\boldsymbol{z}_k)] \right],$$

$$\tag{10}$$

$$- \mathbb{E}_{p_D(\boldsymbol{x})}[D_{KL}(q_\theta(\boldsymbol{z}|\boldsymbol{x})\|r(\boldsymbol{z}))] + \mathbb{E}_{q_\theta(\boldsymbol{x}|\boldsymbol{z})}[[D_{KL}(q_\theta(\boldsymbol{z})\|r(\boldsymbol{z}))],$$

$$\geq \sum_{k=1}^{K} \left[ H(\boldsymbol{x}) + \mathbb{E}_{q_\theta(\boldsymbol{z}_k, \boldsymbol{x})}[\log p_\phi(\boldsymbol{x}|\boldsymbol{z}_k)] - \mathbb{E}_{p_D(\boldsymbol{x})}[D_{KL}(q_\theta(\boldsymbol{z}|\boldsymbol{x})\|r(\boldsymbol{z}))], \right.$$

$$= \sum_{k=1}^{K} \left[ H(\boldsymbol{x}) + \int \left( \int q_\theta(\boldsymbol{z}, \boldsymbol{x}) d\boldsymbol{z}_{\neq k} \right) \log p_\phi(\boldsymbol{x}|\boldsymbol{z}_k) d\boldsymbol{z}_k d\boldsymbol{x} \right] - \mathbb{E}_{p_D(\boldsymbol{x})}[D_{KL}(q_\theta(\boldsymbol{z}|\boldsymbol{x})\|r(\boldsymbol{z}))],$$

$$= \sum_{k=1}^{K} \left[ H(\boldsymbol{x}) + \mathbb{E}_{q_\theta(\boldsymbol{z}, \boldsymbol{x})}[\log p_\phi(\boldsymbol{x}|\boldsymbol{z}_k)] - \mathbb{E}_{p_D(\boldsymbol{x})}[D_{KL}(q_\theta(\boldsymbol{z}|\boldsymbol{x})\|r(\boldsymbol{z}))], \right.$$

$$= KH(\boldsymbol{x}) + \mathbb{E}_{q_\theta(\boldsymbol{z}, \boldsymbol{x})}[\log \prod_{k=1}^{K} p_\phi(\boldsymbol{x}|\boldsymbol{z}_k)] - \mathbb{E}_{p_D(\boldsymbol{x})}[D_{KL}(q_\theta(\boldsymbol{z}|\boldsymbol{x})\|r(\boldsymbol{z}))],$$

$$= KH(\boldsymbol{x}) + \mathbb{E}_{q_\theta(\boldsymbol{z}, \boldsymbol{x})}[\log p_\phi(\boldsymbol{x}|\boldsymbol{z}) + \log p_D(\boldsymbol{x})^{K-1}] - \mathbb{E}_{p_D(\boldsymbol{x})}[D_{KL}(q_\theta(\boldsymbol{z}|\boldsymbol{x})\|r(\boldsymbol{z}))],$$

$$= \mathbb{E}_{q_\theta(\boldsymbol{z}|\boldsymbol{x})}[\log p_\phi(\boldsymbol{x}|\boldsymbol{z})] - \underbrace{\mathbb{E}_{p_D(\boldsymbol{x})}[D_{KL}(q_\theta(\boldsymbol{z}|\boldsymbol{x})\|r(\boldsymbol{z}))]}_{\text{VIB}} =: \mathcal{L}(\boldsymbol{z}, \boldsymbol{x}).$$

Maximizing $\mathcal{L}(\boldsymbol{z}, \boldsymbol{x})$ not only maximizes the original objective $TC(\boldsymbol{z}, \boldsymbol{x})$, but at the same time minimize the gap produced by upper bounding equation 10. As a result,

$$\sum_{k=1}^{K} \left[ \mathbb{E}_{q_\theta(\boldsymbol{z}_k)}[D_{KL}(q_\theta(\boldsymbol{x}|\boldsymbol{z}_k)\|p_\phi(\boldsymbol{x}|\boldsymbol{z}_k))] \right] + \mathbb{E}_{q_\theta(\boldsymbol{x}|\boldsymbol{z})}[[D_{KL}(q_\theta(\boldsymbol{z})\|r(\boldsymbol{z}))], \tag{11}$$

will be minimized, leading to: $r(\boldsymbol{z}) \approx q_\theta(\boldsymbol{z})$ and $p_\phi(\boldsymbol{x}|\boldsymbol{z}_k) \approx q_\theta(\boldsymbol{x}|\boldsymbol{z}_k)$.

Moreover, since $H(\boldsymbol{x})$ and $\log p_D(\boldsymbol{x})^{K-1}$ do not depend on $\theta$, we can drop them from $\mathcal{L}(\boldsymbol{z}, \boldsymbol{x})$. Finally, to avoid using a heavy notation, we will denote the VIB term as $D_{KL}(q_\theta(\boldsymbol{z}|\boldsymbol{x})\|r(\boldsymbol{z}))$, leading to the first TC bound which introduces the VIB term:

$$TC_\theta(\boldsymbol{z}, \boldsymbol{x}) \geq \mathbb{E}_{q_\theta(\boldsymbol{z}|\boldsymbol{x})}[\log p_\phi(\boldsymbol{x}|\boldsymbol{z})] - \underbrace{D_{KL}(q_\theta(\boldsymbol{z}|\boldsymbol{x})\|r(\boldsymbol{z}))}_{\text{VIB}}. \tag{12}$$

## A.2 TC BOUND AND THE CONDITIONAL VARIATIONAL INFORMATION BOTTLENECK

Expanding Eq. equation 2, we can reformulate $TC(\boldsymbol{z}, \boldsymbol{x})$ as follow:

$$TC_\phi(\boldsymbol{z}, \boldsymbol{x}) = \sum_{k=1}^{K} I_\phi(\boldsymbol{z}_k, \boldsymbol{x}) - I_\phi(\boldsymbol{z}, \boldsymbol{x}), \tag{13}$$

$$= \sum_{k=1}^{K} \left[ \frac{K-1}{K} I_\phi(\boldsymbol{z}_k, \boldsymbol{x}) + \frac{1}{K} I_\phi(\boldsymbol{z}_k, \boldsymbol{x}) - \frac{1}{K} I_\phi(\boldsymbol{z}, \boldsymbol{x}) \right],$$

$$= \sum_{k=1}^{K} \left[ \frac{K-1}{K} I_\phi(\boldsymbol{z}_k, \boldsymbol{x}) + \frac{1}{K} \left( I_\phi(\boldsymbol{z}_k, \boldsymbol{x}) - I_\phi(\boldsymbol{z}, \boldsymbol{x}) \right) \right].$$

$$\tag{14}$$

Interestingly, can write the last term of Eq. equation 14 as:

$$I_\phi(\boldsymbol{z}_k, \boldsymbol{x}) - I_\phi(\boldsymbol{z}, \boldsymbol{x}) = \mathbb{E}_{p_\phi(\boldsymbol{x}, \boldsymbol{z}_k)}\left[\log \frac{p_\phi(\boldsymbol{x}|\boldsymbol{z}_k)}{p_D(\boldsymbol{x})}\right] - \mathbb{E}_{p_\phi(\boldsymbol{x}, \boldsymbol{z})}\left[\log \frac{p_\phi(\boldsymbol{x}|\boldsymbol{z})}{p_D(\boldsymbol{x})}\right], \qquad (15)$$

$$= \int \left(\int p_\phi(\boldsymbol{x}|\boldsymbol{z})p(\boldsymbol{z})d\boldsymbol{z}_{\neq \boldsymbol{z}_k}\right)\log\frac{p_\phi(\boldsymbol{x}|\boldsymbol{z}_k)}{p_D(\boldsymbol{x})}d\boldsymbol{z}_k d\boldsymbol{x},$$

$$- \int p_\phi(\boldsymbol{x}|\boldsymbol{z})p(\boldsymbol{z})\log\frac{p_\phi(\boldsymbol{x}|\boldsymbol{z})}{p_D(\boldsymbol{x})}d\boldsymbol{z}d\boldsymbol{x},$$

$$= \int p_\phi(\boldsymbol{x}|\boldsymbol{z})p(\boldsymbol{z})\log\frac{p_\phi(\boldsymbol{x}|\boldsymbol{z}_k)}{p(\boldsymbol{x}|\boldsymbol{z})}d\boldsymbol{z}d\boldsymbol{x},$$

$$= -\int p_\phi(\boldsymbol{x}|\boldsymbol{z})p(\boldsymbol{z})\log\frac{p(\boldsymbol{x}|\boldsymbol{z})}{p_\phi(\boldsymbol{x}|\boldsymbol{z}_k)}d\boldsymbol{z}d\boldsymbol{x},$$

$$= -\int p_\phi(\boldsymbol{x}|\boldsymbol{z})p(\boldsymbol{z})\log\frac{p(\boldsymbol{x}|\boldsymbol{z}_{\neq k}, \boldsymbol{z}_k)}{p_\phi(\boldsymbol{x}|\boldsymbol{z}_k)}d\boldsymbol{z}d\boldsymbol{x},$$

$$= -I_\phi(\boldsymbol{z}_{\neq k}, \boldsymbol{x}|\boldsymbol{z}_k).$$

We can now write equation 5:

$$TC_\theta(\boldsymbol{z}, \boldsymbol{x}) = \frac{1}{K}\sum_{k=1}^{K}\left[(K-1)I_\theta(\boldsymbol{z}_k, \boldsymbol{x}) - I_\theta(\boldsymbol{z}_{\neq k}, \boldsymbol{x}|\boldsymbol{z}_k)\right].$$

Interestingly, the second IB term in Eq. (8) can now be expressed as multiple conditional MIs between the observation and $K-1$ other latent variables given the k-th latent representation variable, penalizing the extra information of the observation not inferable from the given latent representation variable. Moreover, we can further expand the TC as:

$$TC_\theta(\boldsymbol{z}, \boldsymbol{x}) = \frac{1}{K}\sum_{k=1}^{K}\left[(K-1)I_\theta(\boldsymbol{z}_k, \boldsymbol{x}) - I_\theta(\boldsymbol{z}_{\neq k}, \boldsymbol{x}|\boldsymbol{z}_k)\right], \qquad (16)$$

$$= \frac{1}{K}\sum_{k=1}^{K}\left[(K-1)\left[\mathbb{E}_{q_\theta(\boldsymbol{z}_k, \boldsymbol{x})}\left[\log\frac{q_\theta(\boldsymbol{x}|\boldsymbol{z}_k)}{p_D(\boldsymbol{x})}\frac{p_\phi(\boldsymbol{x}|\boldsymbol{z}_k)}{p_\phi(\boldsymbol{x}|\boldsymbol{z}_k)}\right]\right]\right.$$

$$\left. - \mathbb{E}_{q_\theta(\boldsymbol{x}, \boldsymbol{z})}\left[\log\frac{q_\theta(\boldsymbol{z}|\boldsymbol{x})}{q_\theta(\boldsymbol{z}_k|\boldsymbol{x})}\frac{r_p(\boldsymbol{z}|\boldsymbol{x})}{r_p(\boldsymbol{z}|\boldsymbol{x})}\right] + \mathbb{E}_{q_\theta(\boldsymbol{x}, \boldsymbol{z})}\left[\log q_\theta(\boldsymbol{z}_{\neq k})\right]\right],$$

$$= \frac{1}{K}\sum_{k=1}^{K}\left[(K-1)\left[\mathbb{E}_{q_\theta(\boldsymbol{z}_k, \boldsymbol{x})}\left[\log\frac{q_\theta(\boldsymbol{x}|\boldsymbol{z}_k)}{p_D(\boldsymbol{x})}\frac{p_\phi(\boldsymbol{x}|\boldsymbol{z}_k)}{p_\phi(\boldsymbol{x}|\boldsymbol{z}_k)}\right]\right]\right.$$

$$\left. - \mathbb{E}_{p_D(\boldsymbol{x})}[D_{KL}(q_\theta(\boldsymbol{z}|\boldsymbol{x})\|r_p(\boldsymbol{z}|\boldsymbol{x}))] - \mathbb{E}_{q_\theta(\boldsymbol{x}, \boldsymbol{z})}\left[\log\frac{r_p(\boldsymbol{z}_k|\boldsymbol{x})r_p(\boldsymbol{z}_{\neq k}|\boldsymbol{x})}{q_\theta(\boldsymbol{z}_k|\boldsymbol{x})q_\theta(\boldsymbol{z}_{\neq k})}\right]\right],$$

$$= \frac{K-1}{K}\sum_{k=1}^{K}\left[H(\boldsymbol{x}) + \mathbb{E}_{q_\theta(\boldsymbol{z}_k, \boldsymbol{x})}[\log p_\phi(\boldsymbol{x}|\boldsymbol{z}_k)]\right] - \frac{1}{K}\sum_{k=1}^{K}\left[\mathbb{E}_{p_D(\boldsymbol{x})}[D_{KL}(q_\theta(\boldsymbol{z}|\boldsymbol{x})\|r_p(\boldsymbol{z}|\boldsymbol{x}))]\right]$$

$$+ \frac{K-1}{K}\sum_{k=1}^{K}\left[\mathbb{E}_{q_\theta(\boldsymbol{z}_k)}[D_{KL}(q_\theta(\boldsymbol{x}|\boldsymbol{z}_k)\|p_\phi(\boldsymbol{x}|\boldsymbol{z}_k))]\right] + \frac{1}{K}\sum_{k=1}^{K}\mathbb{E}_{q_\theta(\boldsymbol{z}_{\neq k}, \boldsymbol{x})}[D_{KL}(q_\theta(\boldsymbol{z}_k|\boldsymbol{x})\|r_p(\boldsymbol{z}_k|\boldsymbol{x})]$$

$$\tag{17}$$

$$+ \int D_{KL}(q_\theta(\boldsymbol{z}_{\neq k})\|r_p(\boldsymbol{z}_{\neq k}|\boldsymbol{x}))d\boldsymbol{x},$$

$$\geq \frac{K-1}{K}\sum_{k=1}^{K}\left[H(\boldsymbol{x}) + \mathbb{E}_{q_\theta(\boldsymbol{z}_k|\boldsymbol{x})}[\log p_\phi(\boldsymbol{x}|\boldsymbol{z}_k)]\right] - \underbrace{\frac{1}{K}\sum_{k=1}^{K}\left[\mathbb{E}_{p_D(\boldsymbol{x})}[D_{KL}(q_\theta(\boldsymbol{z}|\boldsymbol{x})\|r_p(\boldsymbol{z}|\boldsymbol{x}))]\right]}_{\text{CEB}}.$$

$$\tag{18}$$

Maximizing Eq. 18 not only maximizes the original objective $TC(\boldsymbol{z}, \boldsymbol{x})$ but at the same time minimizes the gap produced by upper bounding Eq. equation 17, leading to: $r_p(\boldsymbol{z}_k|\boldsymbol{x}) \approx q_\theta(\boldsymbol{z}_k|\boldsymbol{x})$, $q_\theta(\boldsymbol{z}_{\neq k}) \approx r_p(\boldsymbol{z}_{\neq k}|\boldsymbol{x})$ and $q_\theta(\boldsymbol{x}|\boldsymbol{z}_k) \approx p_\phi(\boldsymbol{x}|\boldsymbol{z}_k)$. Moreover, since $H(\boldsymbol{x})$ does not depend on $\theta$, we can drop it from Eq. equation 18. Finally, to avoid using a heavy notation, we will denote the CEB term as $D_{KL}(q_\theta(\boldsymbol{z}_k|\boldsymbol{x})\|r_p(\boldsymbol{z}|\boldsymbol{x}))$, leading to the second TC bound which introduces the CEB term:

$$TC_\theta(\boldsymbol{z}, \boldsymbol{x}) \geq \frac{K-1}{K} \mathbb{E}_{q_\theta(\boldsymbol{z}|\boldsymbol{x})}[\log p_\phi(\boldsymbol{x}|\boldsymbol{z})] - \underbrace{D_{KL}(q_\theta(\boldsymbol{z}|\boldsymbol{x})\|r_p(\boldsymbol{z}|\boldsymbol{x}))}_{\text{CEB}}. \tag{19}$$

## A.3 FINAL TC BOUND

In order to obtain the final expression of the derived TC bound, we can compute a convex combination of the two bounds defined in Eq. equation 12 and equation 19.

$$TC(\boldsymbol{z}, \boldsymbol{x}) = (1-\alpha)\left(\sum_{k=1}^{K} I_\theta(\boldsymbol{z}_k, \boldsymbol{x}) - I_\theta(\boldsymbol{z}, \boldsymbol{x})\right) \tag{20}$$

$$+ \alpha\left(\sum_{k=1}^{K}\left[\frac{K-1}{K}I_\theta(\boldsymbol{z}_k, \boldsymbol{x}) + \frac{1}{K}I_\theta(\boldsymbol{z}_k, \boldsymbol{x}) - \frac{1}{K}I_\theta(\boldsymbol{z}, \boldsymbol{x})\right]\right), \tag{21}$$

$$= \frac{K(1-\alpha) + \alpha(K-1)}{K}\sum_{k=1}^{K} I_\theta(\boldsymbol{z}_k, \boldsymbol{x}) - \frac{\alpha}{K}\sum_{k=1}^{K}(I_\theta(\boldsymbol{z}, \boldsymbol{x}) - I_\theta(\boldsymbol{z}_k, \boldsymbol{x})) - (1-\alpha)I_\theta(\boldsymbol{z}, \boldsymbol{x}),$$

$$\geq \frac{K-\alpha}{K}\mathbb{E}_{q_\theta(\boldsymbol{z}|\boldsymbol{x})}[\log p_\phi(\boldsymbol{x}|\boldsymbol{z})] - \alpha D_{KL}(q_\theta(\boldsymbol{z}|\boldsymbol{x})\|r_p(\boldsymbol{z}|\boldsymbol{x})) - (1-\alpha)D_{KL}(p_\theta(\boldsymbol{z}|\boldsymbol{x})\|r(\boldsymbol{z})),$$

$$= \mathbb{E}_{q_\theta(\boldsymbol{z}|\boldsymbol{x})}[\log p_\phi(\boldsymbol{x}|\boldsymbol{z})] - \underbrace{\frac{K\alpha}{K-\alpha}D_{KL}(q_\theta(\boldsymbol{z}|\boldsymbol{x})\|r_p(\boldsymbol{z}|\boldsymbol{x}))}_{\text{CEB}} - \underbrace{\frac{(1-\alpha)}{\left(1-\frac{\alpha}{K}\right)}D_{KL}(q_\theta(\boldsymbol{z}|\boldsymbol{x})\|r(\boldsymbol{z}))}_{\text{VIB}}.$$

where $\alpha$ is a hyperparameter that balances the effects of VIB and CEB terms. Table A.3 illustrates the lower bounds defined for $\beta$-VAE Higgins et al. (2016), FactorVAE Kim & Mnih (2018), HFS Roth et al. (2023) and $\beta$-TCVAE Chen et al. (2018) comparing them to the derived TC bound. We can see that the three bounds present a similar structure, presenting a marginal log-likelihood term and either one or two KL regularizers that impose some kind of information bottleneck.

| Model | Lower Bound |
|-------|-------------|
| $\beta$-VAE | $\mathbb{E}_{q_\theta(\boldsymbol{z}\|\boldsymbol{x})}[\log p_\phi(\boldsymbol{x}\|\boldsymbol{z})] - \beta D_{KL}(q_\theta(\boldsymbol{z}\|\boldsymbol{x})\|p(\boldsymbol{z}))$ |
| FactorVAE | $\mathbb{E}_{q_\theta(\boldsymbol{z}\|\boldsymbol{x})}[\log p_\phi(\boldsymbol{x}\|\boldsymbol{z})] - \beta D_{KL}(q_\theta(\boldsymbol{z}\|\boldsymbol{x})\|p(\boldsymbol{z})) - \gamma D_{KL}(q_\theta(\boldsymbol{z})\| \prod_{k=1}^{K} q_\phi(\boldsymbol{z}_k))$ |
| $\beta$-TCVAE | $\mathbb{E}_{q(z\|n)p(n)}[\log p(n\|z) - \alpha I_q(z;n) - \beta D_{KL}(q(z)\| \prod_j q(\boldsymbol{z}_j)) - \gamma \sum_j D_{KL}(q(z_j)\|p(z_j))$ |
| HFS | $\mathbb{E}_{q_\theta(\boldsymbol{z}\|\boldsymbol{x})}[\log p_\phi(\boldsymbol{x}\|\boldsymbol{z})] - \gamma[\sum_{i=1}^{K-1} \sum_{j=i+1}^{K} \max_{z \in Z_{:,1} \times Z_{:,2} \times \cdots \times Z_{:,K}} \min_{z' \in Z_{:,(i,j)}} d(z,z')]$ |
| $\alpha$-TCVAE | $\mathbb{E}_{q_\theta(\boldsymbol{z}\|\boldsymbol{x})}[\log p_\phi(\boldsymbol{x}\|\boldsymbol{z})] - \frac{K\alpha}{K-\alpha} D_{KL}(q_\theta(\boldsymbol{z}\|\boldsymbol{x})\|r_p(\boldsymbol{z}\|\boldsymbol{x})) - \frac{(1-\alpha)}{(1-\frac{\alpha}{K})} D_{KL}(q_\theta(\boldsymbol{z}\|\boldsymbol{x})\|r(\boldsymbol{z}))$ |

Table 1: This table compares the lower bound objective functions of $\beta$-VAE, $\beta$-TCVAE, Factor-VAE and HFS-VAE. The lower bound objective function of $\beta$-VAE is composed of the expected log-likelihood of the data given the latent variables and the KL divergence between the approximate posterior and the prior of the latent variables (i.e., VIB term). The FactorVAE model further adds a KL divergence term between the approximate posterior and the factorized prior of the latent variables, which approximates the total correlation of the latent variables, and HFS-VAE further adds a Monte-Carlo approximation of Hausdorff distance. $\alpha$-TCVAE, on the other hand, uses a convex combination of VIB term and KL divergence between the approximate posterior and the prior of the latent variables conditioned on the k-th latent variable (i.e., CEB term). $K$ represents the dimensionality of the latent variables, while $\beta$, $\gamma$ and $\alpha$ are hyperparameters of the models.

## B  ARCHITECTURES AND HYPERPARAMETERS DETAILS

The hyperparameters used for the different experiments are shown in Table 2.

Table 2: Comparison of the different hyperparameters used across the Datasets

| Dataset | $\beta$ | $\gamma$ | $\alpha$ | latent dim K | Training Epochs |
|---------|---------|----------|----------|--------------|-----------------|
| Teapots | 2 | 10 | 0.25 | 10 | 50 |
| 3DShapes | 3 | 10 | 0.25 | 10 | 50 |
| Cars3D | 4 | 10 | 0.25 | 10 | 50 |
| MPI3D-Real | 5 | 10 | 0.25 | 10 | 50 |
| Celeba | 5 | 10 | 0.25 | 48 | 50 |

All encoder, decoder and discriminator architectures are taken from Roth et al. (2023).

## C  FURTHER DETAILS ON DATASETS AND METRICS

### C.1  DATASETS

We test on five datasets. **Teapots** (Moreno et al., 2016) contains $200,000$ images of size $64 \times 64$. Each image features a rendered, camera-centered teapot with 5 uniformly distributed generative factors of variation: azimuth and elevation (sampled between 0 and $2\pi$), along with three RGB colour channels (each sampled between 0 and 1). **3DShapes** (Burgess & Kim, 2018) consists of $480,000$ images of size $64 \times 64$. Every image displays a rendered, camera-centered object with 6 uniformly distributed generative factors of variation: shape (sampled from [cylinder, tube, sphere, cube]), object colour, object hue, floor colour, wall colour, and horizontal orientation, all determined using linearly spaced values. **MPI3D-Real** (Gondal et al., 2019) comprises $103,680$ images of size $64 \times 64$. Each image captures objects at a robot arm's end, characterized by 6 factors: object colour, size, shape, camera height, azimuth, and robot arm altitude. **Cars3D** (Reed et al., 2015) is made up of $16,185$ images of size $64 \times 64$. Each image portrays a rendered, camera-centered car, categorized by 3 factors: car-type, elevation, and azimuth. **CelebA** (Liu et al., 2015) encompasses over $200,000$ images of size $64 \times 64$. Every image presents a celebrity, highlighted by a broad range of poses, facial expressions, and lighting conditions, which sum up to 40 different factors. Every model is trained using a subset containing the $80\%$ of the selected dataset images in a fully unsupervised way. The models are evaluated on the remaining images using the following downstream scores. While

CelebA is the most complex dataset, MPI3D-Real is the most realistic among the ones usually used in the disentanglement community.

## C.2 METRICS

When using the **FID** score to assess image quality, we compare the distribution of generated images to that of the real images. Specifically, FID (Heusel et al., 2017) measures the distance between two distributions of images, and we apply it to measure the distance between the generated images and the real ones. A lower distance is better, indicating that the generated images belong to the distribution of ground truth images.

The **Vendi** score (Friedman & Dieng, 2022), which we use to measure the diversity of the generated images, is computed with respect to a similarity measure. Specifically, it is calculated as the exponential of the entropy of the eigenvalues of the similarity matrix, i.e. the matrix whose $(i, j)$th entry is the similarity between the $i$th and $j$th data points. It can be interpreted as the effective number of distinct elements in the set.

To assess the quality of encoded latent representations, we use DCI, SNC/NK (Mahon et al., 2023) and the unfairness measure of Locatello et al. (2019a).

**DCI**, the first disentanglement metric we compute, first trains a regressor to predict the generative factors from the latent representation, and from this regressor extracts a matrix of feature importances, where the $(i, j)$th entry is the import of the $i$th latent dimension to predicting the $j$th generative factor. It then takes (a normalized version of) the entropy of rows and columns to compute 'disentanglement' and 'completeness', respectively. The accuracy of the regressor is taken as the 'informativeness' score. The average of these three scores, across all factors and neurons, is the final DCI score.

**SNC/NK**, the second disentanglement metric we compute, works by first aligning neurons to latent factors using the Kuhn-Munkres algorithm to enforce uniqueness. Then each aligned neuron is used as a classifier for the corresponding factor, by binning its values. A higher accuracy of this single-neuron classifier (SNC) is better, indicating that the factor is well-represented by a single unique neuron. Neuron knockout (NK) is calculated as the difference between an MLP classifier that predicts the generative factor from all neurons, and one that predicts using all neurons but the one that factor was aligned to. A high NK is also better, indicating that no neurons, other than the one it was aligned to, contain information about the given factor. SNC/NK measures a slightly different and stronger notion of disentanglement than DCI, as it explicitly assumes an inductive bias that enforces each factor to be represented by a single latent variable.

**MIG**, is a disentanglement metric that quantifies the degree of separation between the latent variables and the generative factors in a dataset. It calculates the mutual information between each latent variable and each generative factor, identifying the variable that shares the most information with each factor. The gap, or difference, in mutual information between the top two variables for each generative factor is then computed. A larger gap indicates that one latent variable is significantly more informative about a generative factor than the others, signifying a higher degree of disentanglement. This metric is particularly useful in scenarios where a clear and distinct representation of generative factors is desired in the latent space. MIG thus complements DCI and SNC/NK by providing a measure of how well-separated the representations of different generative factors are within the model's latent space.

## D EXTENDED RESULTS

Here, we present further results, in addition to those from Section 4. Figure 10 extended Fig. 7, reporting the correlations also with SNC, NK and the attribute classification accuracy as shown in Figure 8. Unsurprisingly, there is a strong correlations between the three metrics designed to measure disentanglement: DCI, SNC and NK. This, to some extent, verifies the reliability of these different disentanglement metrics. SNC and NK also correlate strongly with Vendi, as DCI does. This further supports the finding in our paper of a relationship between disentanglement and diversity.

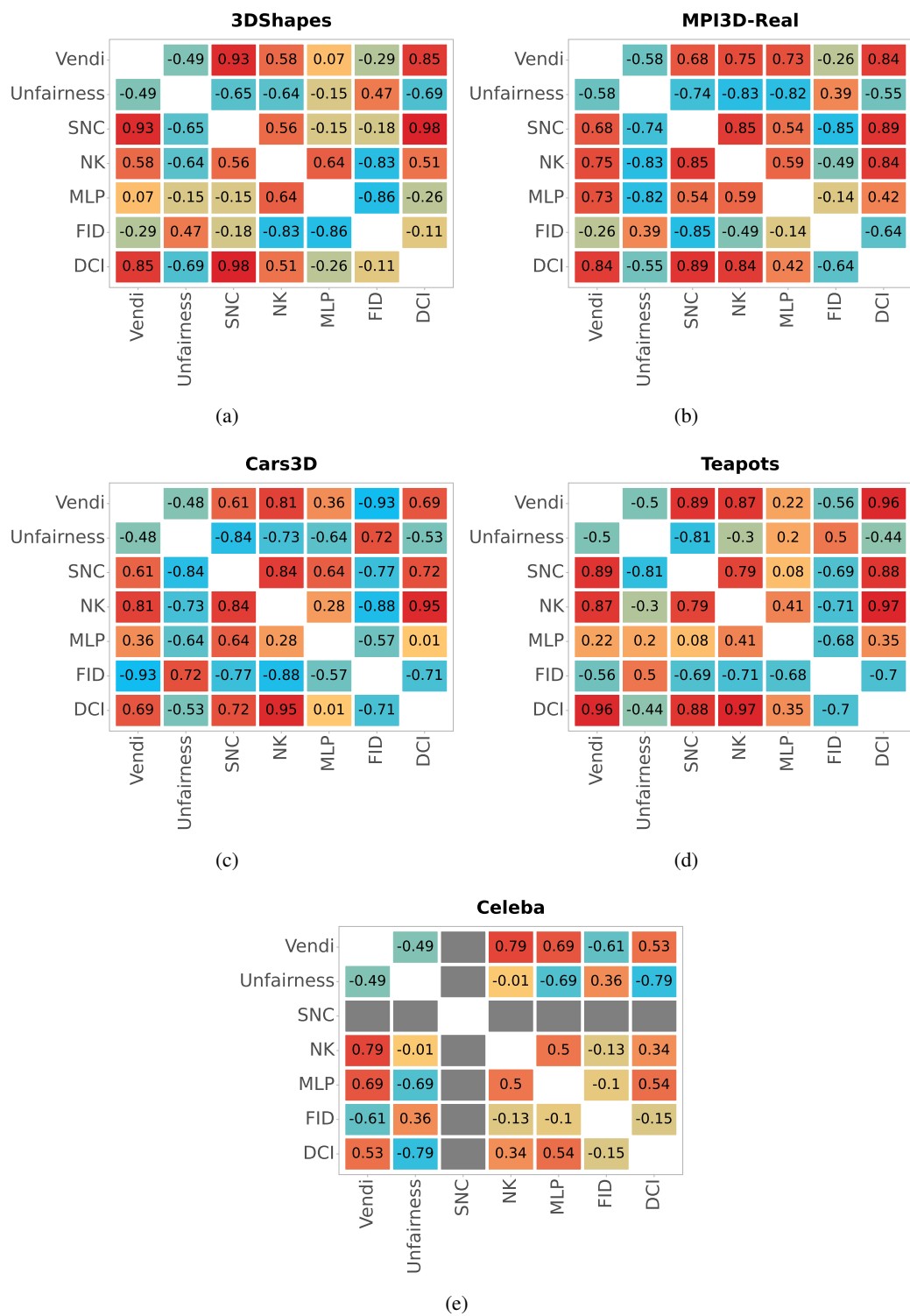

Figure 10: Correlations between all metrics we measure, both for the generated images and the representations.

Figure 11 shows the results for neuron knockout (NK), the second metric introduced by Mahon et al. (2023) alongside SNC, which is shown in Figure 5. Similar to SNC, the NK score for $\alpha$-TCVAE

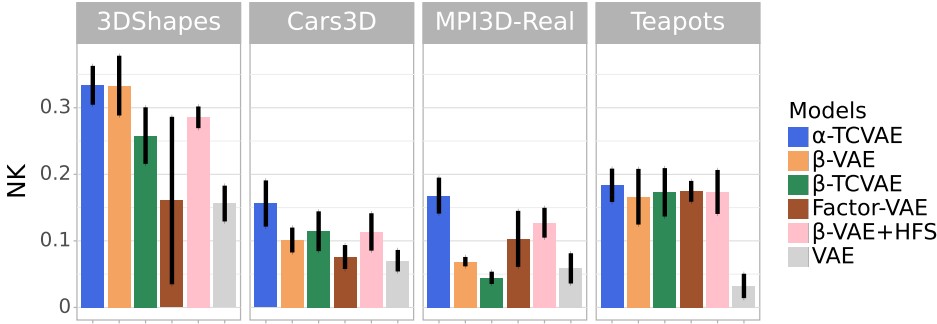

Figure 11: Comparison of the neuron-knockout score of $\alpha$-TCVAE with that of baseline models. As with other metrics presented in the main paper, the improvement of $\alpha$-TCVAE is minor on 3DShapes and Teapots, but more substantial on Cars3D and MPI3D-Real.

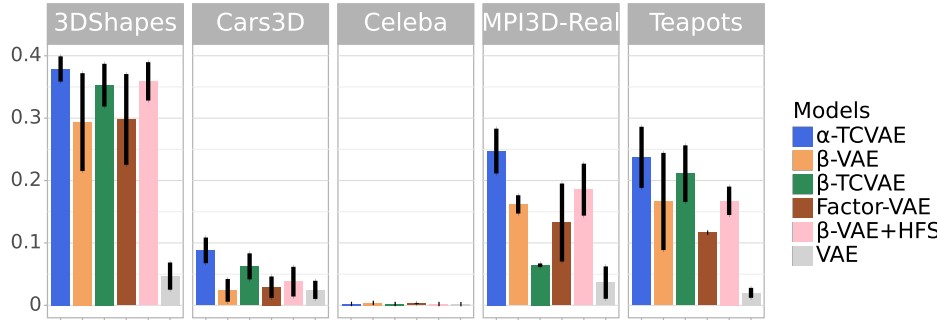

Figure 12: Comparison of the MIG score of $\alpha$-TCVAE with that of baseline models. As with other metrics presented in the main paper, the improvement of $\alpha$-TCVAE is minor on 3DShapes and Teapots, but more substantial on Cars3D and MPI3D-Real.

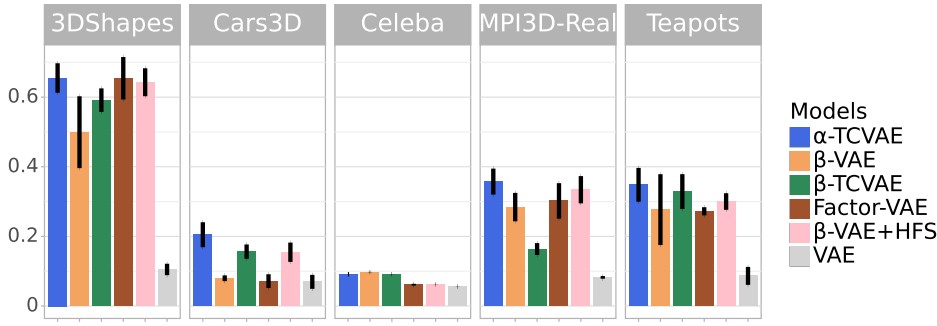

Figure 13: Comparison of the DCI-C completeness score of $\alpha$-TCVAE with that of baseline models. As with other metrics presented in the main paper, the performance of $\alpha$-TCVAE is comparable on 3DShapes, CelebA and Teapots, and better on Cars3D and MPI3D-Real.

is higher than that for baseline VAE models and, while the errorbars often overlap, the superiority of $\alpha$-TCVAE is consistent across all five datasets and is most substantial on MPI3D-Real. Figure 12 shows the results for mutual information gap (MIG), which follows the same trend of NK, SNC and DCI scores. Figures 13, 14 and 15 present the results of Completeness, Disentanglement and

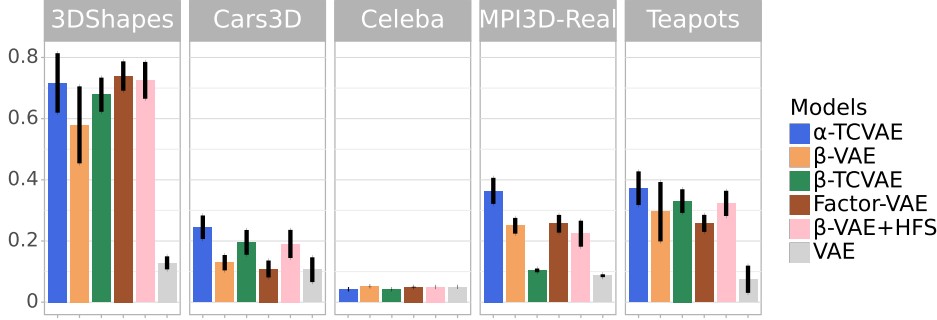

Figure 14: Comparison of the DCI-D disentanglement score of $\alpha$-TCVAE with that of baseline models. As with other metrics presented in the main paper, the performance of $\alpha$-TCVAE is comparable on 3DShapes, CelebA and Teapots, and better on Cars3D and MPI3D-Real.

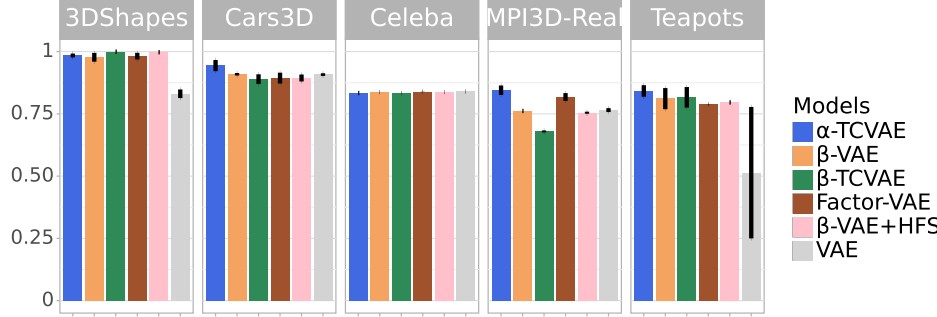

Figure 15: Comparison of the DCI-I informativeness score of $\alpha$-TCVAE with that of baseline models. As with other metrics presented in the main paper, the performance of $\alpha$-TCVAE is comparable on 3DShapes, CelebA and Teapots, and better on Cars3D and MPI3D-Real.

Informativeness metrics (DCI-C, DCI-D and DCI-I, respectively). The final DCI scores shown in fig. 4 is computed as geomtric mean of the three scores.

## E  DISCOVERING NOVEL FACTOR OF VARIATIONS

Figure 16 presents $\alpha$-TCVAE traversals across 3DShapes, Teapots and MPI3D-Real datasets. The red boxes indicate the discovered novel generative factors, that are not present within the train dataset, namely object position and vertical camera perspective. While we do not have a comprehensive explanation of why such an intriguing phenomenon is shown, we believe that the intuition behind can be explained considering the effects of VIB and CEB terms in the defined bound. Indeed, while VIB pushes individual latent variables to represent different generative factors, CEB pushes them to be informative. As a result, the otherwise noisy dimensions, are pushed to be informative (i.e., CEB) and to represent a distinct generative factor (i.e., VIB), resulting in the discovery of novel generative factors.

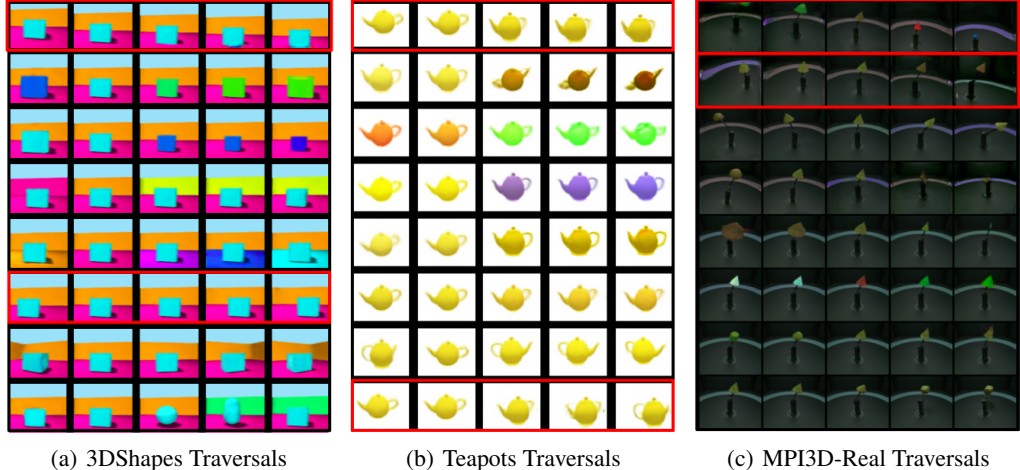

(a) 3DShapes Traversals          (b) Teapots Traversals          (c) MPI3D-Real Traversals

Figure 16: $\alpha$-TCVAE generated latent traversals the 3DShapes, Teapots and MPI3D-Real datasets. The generated latent traversals reveal that $\alpha$-TCVAE can learn and represent generative factors that are not present in the ground-truth dataset, namely vertical perspective and object position. The discovered generative factors are indicated with a red box.

## F  RELATIONSHIP BETWEEN CEB AND DIVERSITY

### F.1  FISHER'S DEFINITION OF CONDITIONAL ENTROPY BOTTLENECK

Fisher's approach to the Conditional Entropy Bottleneck Fischer & Alemi (2020) is an extension of the Information Bottleneck (IB) principle Alemi et al. (2017), aimed at finding an optimally compressed representation of a variable $X$ that remains highly informative about another variable $Y$, under the influence of a conditioning variable $Z$. The CEB objective, according to Fisher, is formalized as a trade-off between two competing conditional mutual information terms:

$$\min_{p(z|x)} \left[ I(X;Z|C) - \beta I(Y;Z|C) \right]$$

Here, $I(X;Z|C)$ quantifies the amount of information that the representation $Z$ shares with $X$, conditioned on $C$. Simultaneously, $I(Y;Z|C)$ measures how much information $Z$ retains about $Y$, also under the condition of $C$. The parameter $\beta$ serves as a crucial tuning parameter, balancing these two aspects.

### F.2  ADAPTING CEB TO VAES WITHOUT CONDITIONING VARIABLES

In the realm of Variational Autoencoders, where the training strategy is to reconstruct the input data $X$ using a latent representation $Z$ without any external conditioning C, the CEB framework

undergoes a significant simplification. Given that $X = Y$ in a typical VAE setup, the CEB objective reduces to a form where the focus shifts to optimizing the mutual information between $X$ and its latent representation $Z$:

$$\min_{p(z|x)} [(1 - \beta)I(X; Z)]$$

This objective can be further broken down as $(1 - \beta)(H(X) - H(X|Z))$, where $H(X)$ represents the entropy of the input data, and $H(X|Z)$ is the conditional entropy of the input given its latent representation. This formulation underscores the trade-off between compressing the input data in the latent space and retaining essential information for accurate reconstruction.

### F.3    INCORPORATING DIVERSITY INTO THE CEB OBJECTIVE

Following Friedman & Dieng (2022), Diversity can be quantitatively expressed as the exponential of the entropy of the latent space distribution $q(Z|X)$:

$$\text{Diversity} = \exp(H(Z|X))$$

To understand how the CEB framework relates to this notion of diversity, we utilize the entropy chain rule $\mathrm{H}(Y|X) = \mathrm{H}(X, Y) - \mathrm{H}(X)$ , which allows to decompose $H(X|Z)$ in terms of the joint entropy $H(X, Z)$ and the conditional entropy $H(Z)$. Consequently, the CEB objective evolves into a more comprehensive form that explicitly accounts for the diversity of the latent space:

$$\min_{q(z|x)} [(1 - \beta)(H(X) - H(X, Z) + H(Z))]$$

$$\min_{q(z|x)} [(1 - \beta)(-H(Z|X) + H(Z))]$$

The latter one, makes clear the connection between the CEB term and Diversity as defined in Friedman & Dieng (2022). Indeed, we can see that when minimizing the CEB term the Diversity term is maximized.

## G    DISENTANGLEMENT AND VARIATIONAL INFORMATION BOTTLENECK

Disentanglement in VAEs, following Higgings' $\beta$-VAE framework, seeks to learn representations where individual latent variables capture distinct, independent factors of variation in the data. This is achieved by modifying the traditional VAE objective to apply a stronger constraint on the latent space information bottleneck, controlled by a hyperparameter $\beta$. The $\beta$-VAE, introduced by Higgins et al. (2016), represents a seminal approach to disentanglement, promoting the learning of factorized and interpretable latent representations.

On a related front, the Variational Information Bottleneck (VIB) method, formulated by Alemi et al. (2017), extends the Information Bottleneck principle to deep learning. The VIB approach seeks to find an optimal trade-off between the compression of input data and the preservation of relevant information for prediction tasks. By employing a variational approximation, VIB efficiently learns compressed representations that are predictive of desired outcomes. Interestingly, Alemi formulates a VIB objective that is equivalent to Higgins' $\beta$-VAE one. Such result makes evident how imposing a higher information bottleneck leads to higher disentanglement.

## H    SENSITIVITY ANALYSIS OF $\alpha$

In this section we present a sensitivity analysis of how $\alpha$ affects Vendi and FID results across the considered datasets. To be consistent and analyze how Alpha influences disentanglement scores, we also report a sensitivity analysis of the DCI metric and a correlation study showing how alpha is statistically correlated with FID, Vendi, and DCI metrics.

## H.1 DIVERSITY AND VISUAL FIDELITY SENSITIVITY ANALYSIS WITH RESPECT TO $\alpha$

To analyse how $\alpha$ influences the presented results, we performed an evaluation of FID, Vendi and DCI using $\alpha \in [0.00, 0.25, 0.50, 0.75, 1.00]$, where for $\alpha = 0.00$ we obtain $\beta$-VAE model, while for $\alpha = 0.25$ we get the results presented in the main paper. Figures 17 and 18 show that, when $\alpha \in [0.25, 0.50]$ $\alpha$-TCVAE presents the highest diversity scores, while keeping a FID score comparable to $\beta$-VAE.

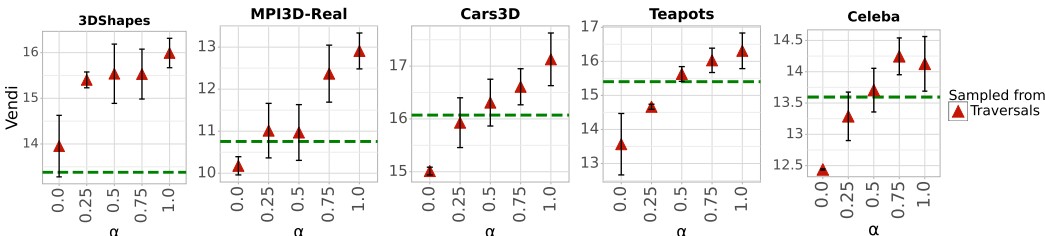

Figure 17: Sensitivity Analysis of the Diversity of generated images with respect to $\alpha$. Only one sampling strategy is considered: sampled from traversals. The green dashed line represents ground truth dataset diversity. It can be seen that the higher alpha the higher will be the Vendi Score.

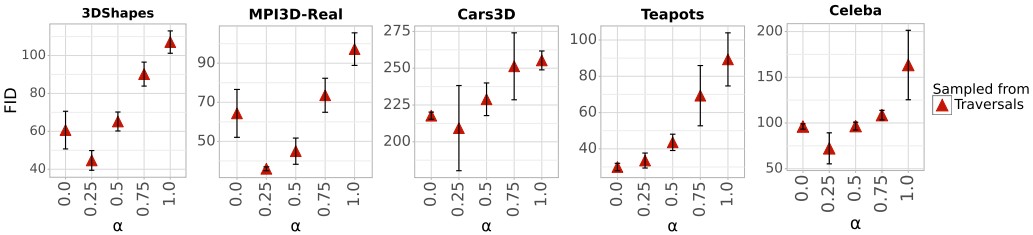

Figure 18: Sensitivity Analysis of the Faithfulness of generated images to the data distribution, as measured by FID score, with respect to $\alpha$. Only one sampling strategy is considered: sampled from traversals. It can be seen that for $\alpha \in [0.25, 0.50]$ the model presents higher visual fidelity.

Interestingly, the two sensitivity analyses show two main trends:

- Diversity increases when using higher values of Alpha.

- FID score improves when using smaller values of Alpha.

Indeed, when using higher values of $\alpha$, we increase the contribution of the CEB term in equation 6, which enhances diversity at the cost of visual fidelity. As a result, the higher the value of $\alpha$, the more diverse the generated batch of images, and the lower will be the generation quality. However, it can be noticed that when using values of $\alpha$ between 0.25 and 0.50, we get a set of generated images that are more diverse and still have a better or comparable visual fidelity than $\beta$-VAE (i.e., $\alpha$=0).

## H.2 DISENTANGLEMENT SENSITIVITY ANALYSIS WITH RESPECT TO $\alpha$

Here, we present a sensitivity analysis of the DCI metric. Figure 19 shows that the interval [0.25-0.50] presents higher values of disentanglement, following Diversity and Visual Fidelity analyses that show the best results in the same range. Such a trend can be explained by considering that $\alpha$ weights the contributions of VIB and CEB terms. While the CEB term enhances diversity, the VIB term encourages disentanglement. As a result, we can see that DCI scores decrease when $\alpha$ gets closer to 1. Interestingly, when $\alpha$ is in [0.25,0.50], the combination of CEB and VIB terms produces a better bound for the Total Correlation objective than when using , which results in higher DCI scores.

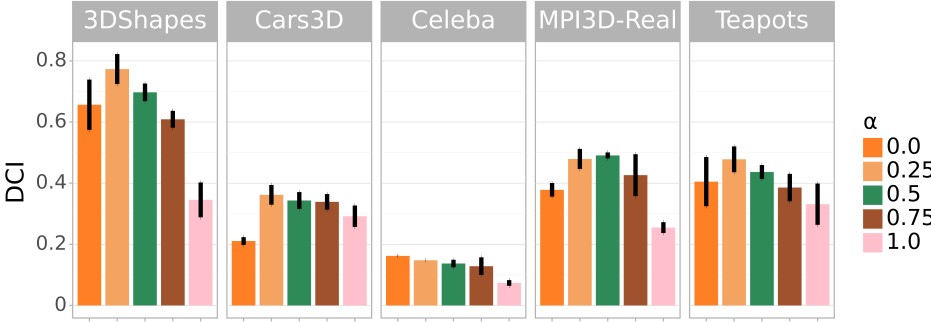

Figure 19: DCI scores sensitivity analysis with respect to $\alpha$. On average when $\alpha \in [0.25, 0.50]$ $\alpha$-TCVAE presents the best DCI scores.

### H.3 CORRELATION STUDY: HOW IS $\alpha$ CORRELATED WITH VENDI, FID, AND DCI METRICS?

Here, we present correlation matrices for all the considered datasets. We computed them using the models trained for the alpha sensitivity analyses. The correlation matrices in fig. 20 confirm the trends observed in the other sensitivity analyses (i.e., Vendi, FID, and DCI). Indeed, $\alpha$ has a strong positive correlation with both FID and Vendi, showing that when $\alpha$ increases, diversity increases and FID deteriorates. On the other hand, $\alpha$ has a strong negative correlation with DCI for all datasets besides the Cars 3D dataset, showing that, on average, the higher the value of $\alpha$, the lower the disentanglement.

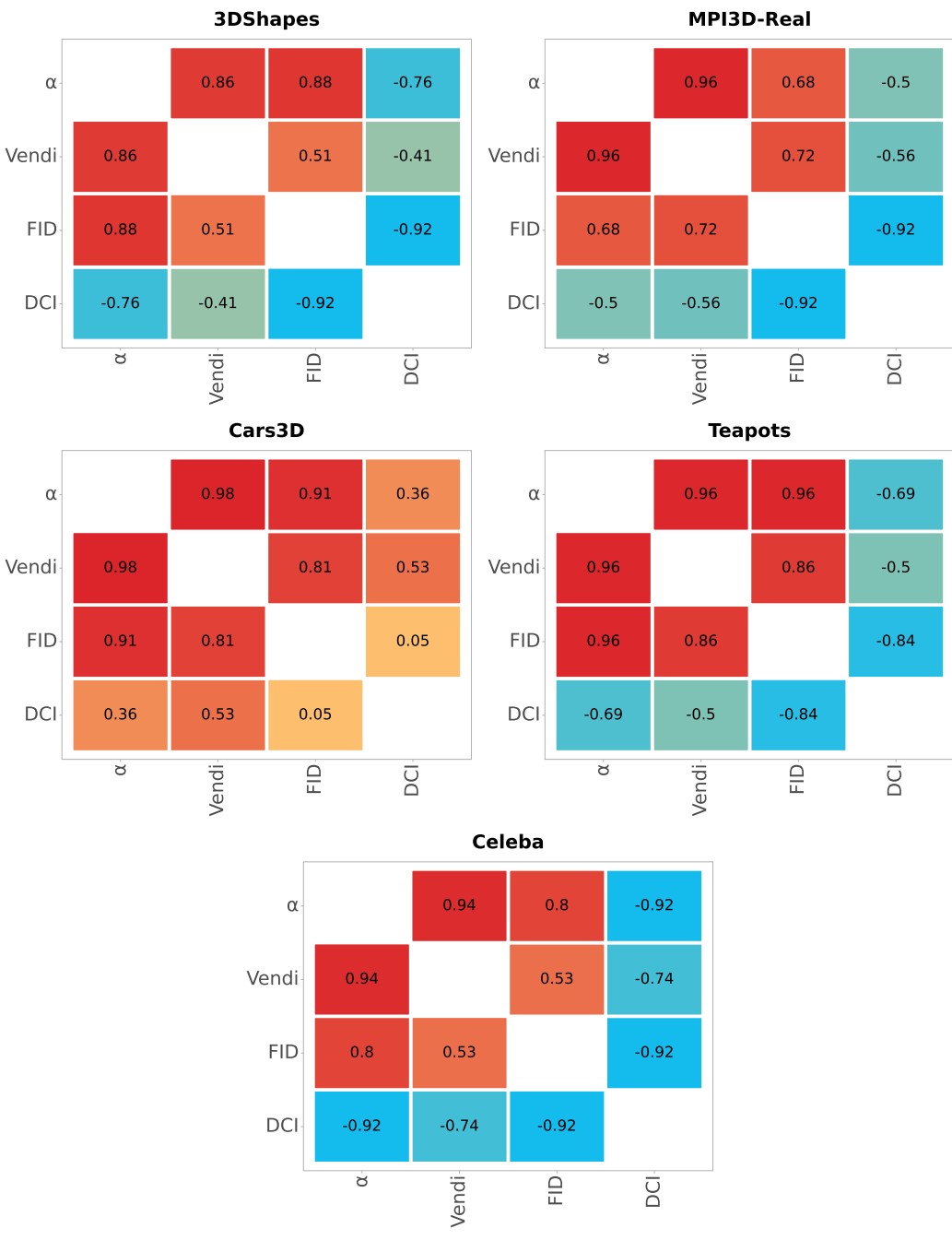

Figure 20: Correlations between $\alpha$, diversity (Vendi score), generation faithfulness (FID score), and disentanglement (DCI). Correlations are computed using the results from all models across 5 different seeds.

