# OpenReview forum: "$\alpha$TC-VAE: On the relationship between Disentanglement and Diversity"
_ICLR.cc/2024/Conference — ICLR 2024 poster_

### Official Review · Reviewer_4hmE · 2023-10-27

**Soundness:** 2 fair
**Presentation:** 3 good
**Contribution:** 2 fair
**Rating:** 5
**Confidence:** 4

**Summary:**

From the perspective of information theory, this paper decomposes the Total Correlation term into two distinct bounds: the information bottleneck and the conditional entropy bottleneck. It assigns specific weights to these bounds to balance their influence. Furthermore, the authors evaluate their learning objective across multiple datasets.

**Strengths:**

- From an information theory perspective, the authors derive a lower bound of total correlation. They simultaneously address both the disentanglement and informativeness of the latent variable.
- The writing throughout the paper is lucid and well-structured, making it accessible for readers.
- The research showcases effectiveness by employing diverse evaluation metrics and testing their approach across various datasets.

**Weaknesses:**

- The novelty is limited. Earlier research, referenced by the authors, already explored the decomposition of Total Correlation. The primary contribution seems to be the weighted combination of these two decompositions, which might appear incremental.
- They claim that the VIB term promotes compression of the latent representation, and the CEB term promotes balance between the information contained in each latent dimension. However, the paper lacks a clear illustration of:
  - How the information bottleneck correlates with disentanglement.
  - The connection between diversity and the conditional entropy bottleneck.
- A more profound point of contention is the origin of the VIB and the CEB terms. Both terms are derived from the same TC term, yet they supposedly represent different characteristics. The paper doesn't sufficiently elucidate why this is the case, which could have made for a compelling discussion.

**Questions:**

- How does the parameter alpha influence the results?
- What about the traversal outcomes for the CelebA dataset?

---

> ### Author Response · Authors · 2023-11-17
> **Answer to Weakness 1**
>
> Q3.1: The novelty is limited. Earlier research, referenced by the authors, already explored the decomposition of Total Correlation. The primary contribution seems to be the weighted combination of these two decompositions, which might appear incremental.
>
> A3.1: Although the decomposition of Total Correlation has been already explored and the final loss is incremental with respect to Beta-VAE loss, no convex lower bounds that can be computed without using extra networks have been proposed so far for the Total Correlation. Not only the proposed bound is grounded on information theory constructs (VIB and CEB terms), but since it doesn’t require the usage of other networks, as in FactorVAE (H. Kim & A. Mnih, 2019) where a discriminator network is required, it can easily be applied on top of other models.
>
> Moreover, while beta-TCVAE (R. Chen et al. 2019) uses a Total Correlation bound as well, they make use of sampling-based strategies to compute the mutual information component used in their loss. However, such sampling-based strategies have several drawbacks. One of the primary disadvantages is that the accuracy of entropy estimation heavily relies on the size of the sample. When the sample size is too small, especially in high-dimensional spaces, it may not represent the true distribution of the data well. This leads to biased or inaccurate estimates of entropy. Furthermore, the accuracy of entropy estimates can be significantly influenced by the sampling method used. If the sampling method introduces biases or fails to capture certain aspects of the underlying distribution, the entropy estimate will be affected. Finally, in high-dimensional spaces, the amount of data required to accurately estimate entropy grows exponentially. This is known as the curse of dimensionality. Sampling-based methods struggle in these scenarios as obtaining a representative sample that captures all the nuances of a high-dimensional distribution is extremely challenging. We would like to clarify that in our case no sampling is required to compute the TC bound, overcoming all the limitations related to sampling-based approaches.

---

> ### Author Response · Authors · 2023-11-17
> **Answer to Weakness 2**
>
> Q3.2: They claim that the VIB term promotes compression of the latent representation, and the CEB term promotes balance between the information contained in each latent dimension. However, the paper lacks a clear illustration of:
> How the information bottleneck correlates with disentanglement.
> The connection between diversity and the conditional entropy bottleneck.
> A3.2:
>
> RELATIONSHIP BETWEEN IB AND Disentanglement
>
> Disentanglement in VAEs, following Higgings’ β-VAE framework, seeks to learn representations where individual latent variables capture distinct, independent factors of variation in the data. This is achieved by modifying the traditional VAE objective to apply a stronger constraint on the latent space information bottleneck, controlled by a hyperparameter β. The β-VAE, introduced by Higgins
> et al. (2016), represents a seminal approach to disentanglement, promoting the learning of factorized and interpretable latent representations.
> On a related front, the Variational Information Bottleneck (VIB) method, formulated by Alemi et al. (2017), extends the Information Bottleneck principle to deep learning. The VIB approach seeks to find an optimal trade-off between the compression of input data and the preservation of relevant information for prediction tasks. By employing a variational approximation, VIB efficiently learns compressed representations that are predictive of desired outcomes. Interestingly, Alemi formulates a VIB objective that is equivalent to Higgins’ β-VAE one. Such a result makes clear the relationship between Information Bottleneck and Disentanglement, concluding that imposing a higher information bottleneck leads to higher disentanglement.
>
>
>
> RELATIONSHIP BETWEEN CEB AND DIVERSITY
>
> FISHER’S DEFINITION OF CONDITIONAL ENTROPY BOTTLENECK
> Fisher’s approach to the Conditional Entropy Bottleneck Fischer & Alemi (2020) is an extension
> of the Information Bottleneck (IB) principle Alemi et al. (2017), aimed at finding an optimally
> compressed representation of a variable X that remains highly informative about another variable
> Y , under the influence of a conditioning variable Z. The CEB objective, according to Fisher, is
> formalized as a trade-off between two competing conditional mutual information terms:
>
> min_q(z|x) [I(X; Z|C) − βI(Y ; Z|C)]
>
> Here, I(X; Z|C) quantifies the amount of information that the representation Z shares with X,
> conditioned on C. Simultaneously, I(Y ; Z|C) measures how much information Z retains about Y ,
> also under the condition of C. The parameter β serves as a crucial tuning parameter, balancing these two aspects.
>
>
> ADAPTING CEB TO VAES WITHOUT CONDITIONING VARIABLES
> In the realm of Variational Autoencoders, where the training strategy is to reconstruct the input
> data X using a latent representation Z without any external conditioning C, the CEB framework
> undergoes a significant simplification. Given that X = Y in a typical VAE setup, the CEB objective
> reduces to a form where the focus shifts to optimizing the mutual information between X and it's
> latent representation Z:
>
> min_q(z|x) [(1 − β)I(X; Z)]
>
> This objective can be further broken down as (1 − β)(H(X) − H(X|Z)), where H(X) represents
> the entropy of the input data, and H(X|Z) is the conditional entropy of the input given its latent
> representation. This formulation underscores the trade-off between compressing the input data in
> the latent space and retaining essential information for accurate reconstruction.
>
> On the relationship between Diversity and CEB
> Following Friedman & Dieng (2022), Diversity can be quantitatively expressed as the exponential
> of the entropy of the latent space distribution q(Z|X):
> Diversity = exp(H(Z|X))
> To understand how the CEB framework relates to this notion of diversity, we utilize the entropy
> chain rule H(Y |X) = H(X, Y ) − H(X) , which allows to decompose H(X|Z) in terms of the
> joint entropy H(X, Z) and the conditional entropy H(Z). Consequently, the CEB objective evolves
> into a more comprehensive form that explicitly accounts for the diversity of the latent space:
>
> min_q(z|x) [(1 − β)(H(X) − H(X, Z) + H(Z))]
>
> min_q(z|x) [(1 − β)(−H(Z|X) + H(Z))]
>
> The latter one, makes clear the connection between the CEB term and Diversity as defined in Fried-
> man & Dieng (2022). Indeed, we can see that when minimizing the CEB term the Diversity term is
> Maximized.

---

> ### Author Response · Authors · 2023-11-17
> **Answer to Weakness 3**
>
> A3.3:
> Appendix A.1 presents the derivation of the proposed TC bound. In order to derive it, we used the same trick - making explicit different quantities from the starting TC expression and upper bounding differently the two expressions - such an approach produces two different upper bounds and two related gaps that will be minimized when optimizing the bound. In appendix A.1 we write:
>
> “Maximizing L(z, x) not only maximizes the original objective TC(z, x), but at the same time
> minimize the gap produced by upper bounding equation 10. As a result, ∑k=1[Eqθ (zk )[DKL(qθ (x|zk)∥pφ(x|zk))]] + Eqθ (x|z)[[DKL(qθ (z)∥r(z))], (11)
> will be minimized, leading to: r(z) ≈ qθ (z) and pφ(x|zk) ≈ qθ (x|zk).”
>
> “Maximizing Eq. 18 not only maximizes the original objective T C(z, x) but at the same time min-
> imizes the gap produced by upper bounding Eq. equation 17, leading to: rp(zk|x) ≈ qθ (zk|x),
> qθ (z̸=k) ≈ rp(z̸=k|x) and qθ (x|zk) ≈ pφ(x|zk)”
>
> Hence, it is important to clarify that the two terms are substantially different because of 1) the expressions used to define the TC and 2) the components dropped to upper bound the Total Correlation.  The gap minimized in the two terms is different, which is why the two bounds end up being different as well.

---

> > ### Comment · Reviewer_4hmE · 2023-11-21
> >
> > Both of these upper bounds are derived from the following term:$ TC_{\theta}({z},{x}) = \sum_{k=1}^{K}I_\theta (z_k, x) - I_\theta (z, x) $. As stated by the authors in the text:
> >    - VIB makes $ r(z) \approx q_\theta (z) $ and $p_\phi(x|z_k) \approx q_\theta (x|z_k) $,
> >    - CEB results in: $r_p(z_k|x) \approx q_\theta (z_k|x)$, $q_\theta (\bar{z}=k) \approx r_p(\bar{z}=k|x) $and$q_\theta (x|z_k) \approx p_\phi(x|z_k)$.
> >
> > The VIB and CEB upper bounds are derived from the same underlying term and share similarities in their optimization outcomes. These shared outcomes imply that effectively optimizing these bounds requires a deep understanding of their interaction and the distinct characteristics each represents with respect to TC. For instance, this may involve sensitive analysis of hyperparameters like $\alpha$, which plays a critical role in balancing the trade-offs inherent in these models.

---

> > > ### Author Response · Authors · 2023-11-22
> > >
> > > Dear reviewer,
> > >
> > > As you suggested, we performed a sensitivity analysis of $\alpha$. We added it in the appendix, together with a section that explains the connection between the CEB term and Diversity. We hope that the given answers, together with the new presented results, are convincing enough to raise the given rating.
> > >
> > > We are available to answer any further question, we thank the reviewer for the provided feedback and the availability.

---

> > > > ### Comment · Reviewer_4hmE · 2023-11-23
> > > > **Response**
> > > >
> > > > Thanks for your response. My comments have been addressed, so I'll bump up the score.

---

> > > > > ### Author Response · Authors · 2023-11-23
> > > > >
> > > > > Thank you so much for your availability and useful feedback!

---

> ### Author Response · Authors · 2023-11-17
> **Answer to Q2**
>
> Q3.4: What about the traversal outcomes for the CelebA dataset?
>
> A3.4: We did not add CelebA traversals because we used 48 dimensional latent space, which makes it a bit impractical producing the traversals and including them in the paper.  Here you can find a link to an anonymous drive that contains the related latent traversals:
> https://drive.google.com/file/d/1V-9bM70z0YwrzpRpiqP8rHk8GGXtx8NY/view?usp=sharing . The presented traversals show more than 15 uncovered generative factors ( which is the highest number of uncovered generative factors in the literature (R. Chen et al. 2019)  for the Celeba dataset). Moreover some of the uncovered generative factors are not present in the training set, for instance the background color (18th row of the traversal).

---

> ### Author Response · Authors · 2023-11-17
>
> We thank the reviewer for the outlined weaknesses. We were aware of some of them, which is why we prepared some of the answers beforehand (i.e., relationship between CEB and Diversity). We are currently preparing the plots for some experiments we perform with the goal of clarifying how alpha influences the results. We hope that together with the previous questions, these answers can lead the reviewer to increase the grade of the review. If there is any other question or point that the reviewer would like to expand on, we are more than available on discussing and answering any kind of question.
>
> Thank you again for your availability!

---

### Official Review · Reviewer_hWoC · 2023-10-29

**Soundness:** 2 fair
**Presentation:** 2 fair
**Contribution:** 2 fair
**Rating:** 5
**Confidence:** 3

**Summary:**

In this paper, the author introduce α-TCVAE, a variational autoencoder optimized using a new total correlation (TC) lower bound that both maximizes disentanglement and latent variables informativeness. The new TC bound is grounded in information theory and can be reduced to a convex combination of the known VIB and CEB terms. Moreover, the paper also presents quantitative analyses and correlation studies which support the idea that smaller latent domains lead to better generative capabilities and diversity.

**Strengths:**

- The total correlation between x and z has not been previously used in disentangled representation learning, although similar concepts exist, such as mutual information in InfoGAN.
- The paper  first paper propose to discuss diversity in the context of disentangled representation learning, drawing from previous research in GANs (e.g., FID).
- The paper considers reinforcement learning as a downstream task for disentangled representations.

**Weaknesses:**

- The paper proposes maximizing an upper bound of the correlation between x and z for disentanglement, but β-TCVAE penalizes total correlation between different z's. The connection between these two approaches is unclear, and the use of α-TCVAE may be misleading. If there is a connection, it should be further clarified.
- The authors should explain why maximizing the upper bound of the correlation between x and z promotes disentanglement, and why increasing variable informativeness enhances diversity.
- The traversal results shown in Appendix Figure 12 are poor in terms of both disentanglement and generation quality. Additionally, the paper lacks traversal images results for real world natural images (not MPI3D-real), such as CelebA.
- The paper should provide an analysis of the sensitivity of the $\alpha$ parameter.
- If the answer to weakness 1 is the latter, it seems that this work is relatively incremental, as it only adds a conditional TC term to the β-TCVAE. Considering this point, the authors should clarify it in the rebuttal stage.

**Questions:**

See weakness.

---

> ### Author Response · Authors · 2023-11-22
>
> We thank the reviewer for the feedback. We updated the manuscript and particularly the appendix to include the material related to these questions.
>
> Q1: The paper proposes maximizing an upper bound of the correlation between x and z for disentanglement, but β-TCVAE penalizes total correlation between different z's. The connection between these two approaches is unclear, and the use of α-TCVAE may be misleading. If there is a connection, it should be further clarified.
>
> A1: The connection between β-TCVAE and α-TCVAE can be explained looking at Equation 2 in the paper:  $TC_{\theta}(\boldsymbol{z}, \boldsymbol{x}) = TC_{\theta}(\boldsymbol{z}) - TC_{\theta} (\boldsymbol{z} | \boldsymbol{x})$.
> There are 2 main differences between β-TCVAE and α-TCVAE:
> 1) As highlighted by the reviewer, while β-TCVAE  tries to optimize $TC_{\theta}(\boldsymbol{z})$, we upper bound  $TC_{\theta}(\boldsymbol{z}, \boldsymbol{x})$ which also takes into account the mutual information shared between latent representation and input space. Indeed, one of the main problems of most of disentangled models is trying to make as much independent as possible the latent representations without taking into account the information that flows from the input space to the latent representation. By contrast, upper bounding  $TC_{\theta}(\boldsymbol{z}, \boldsymbol{x})$ allows our model to better represents and discover more factors of variations in  complex datasets (e.g., MPI3D_Real and CelebA). For instance, while β-TCVAE discovers 15 generative factors in CelebA, we are able to count 25 distinct generative factors in  α-TCVAE (here there is an anonymous link with the generated traversals: https://drive.google.com/file/d/1V-9bM70z0YwrzpRpiqP8rHk8GGXtx8NY/view?usp=sharing ).
>
> 2) While β-TCVAE uses a Monte Carlo sampling approach to compute the mutual information term used in β-TCVAE loss function, we do not need to use any sampling strategy to compute the derived bound. However, such sampling-based strategies have several drawbacks. One of the primary disadvantages is that the accuracy of entropy estimation heavily relies on the size of the sample. When the sample size is too small, especially in high-dimensional spaces, it may not represent the true distribution of the data well. This leads to biased or inaccurate estimates of entropy. Furthermore, the accuracy of entropy estimates can be significantly influenced by the sampling method used. If the sampling method introduces biases or fails to capture certain aspects of the underlying distribution, the entropy estimate will be affected. Finally, in high-dimensional spaces, the amount of data required to accurately estimate entropy grows exponentially. This is known as the curse of dimensionality. Sampling-based methods struggle in these scenarios as obtaining a representative sample that captures all the nuances of a high-dimensional distribution is extremely challenging. We would like to clarify that in our case no sampling is required to compute the TC bound, overcoming all the limitations related to sampling-based approaches.

---

> ### Author Response · Authors · 2023-11-22
>
> Q2: The authors should explain why maximizing the upper bound of the correlation between x and z promotes disentanglement, and why increasing variable informativeness enhances diversity.
>
> We added a section in the Appendix which explains and formally defines the connection between the CEB term and diversity, and the VIB term and disentanglement.
>
> Q3: The traversal results shown in Appendix Figure 12 are poor in terms of both disentanglement and generation quality. Additionally, the paper lacks traversal images results for real world natural images (not MPI3D-real), such as CelebA.
>
> Here there is an anonymous link with the generated traversals for CelebA: https://drive.google.com/file/d/1V-9bM70z0YwrzpRpiqP8rHk8GGXtx8NY/view?usp=sharing . The reason why we did not include them is that since we are using a latent space with 48 dimensions, fitting the image in one page is very unhandy and leads to a very poor image quality.
>
>
> Q4: The paper should provide an analysis of the sensitivity of the parameter.
>
> We added a section in the appendix with the sensitivity analysis of the alpha parameter.
>
> Q5 is explained in the answer we provided for Q1.
>
> We thank the reviewer for the provided feedback. We hope that the sections added in the appendix fully answer the provided feedback and address the highlighted weaknesses, leading to an increase of the rating. We are open to discuss for any further question.

---

> ### Comment · Reviewer_hWoC · 2023-11-23
>
> Thank yor for your response. Some of the concerns are addressed. However, based on your response, I still have the follow two questions:
> 1. What is the difference in KL divergence used in VIB and VAE?
> 2. If I understand correctly, CEB is a diversity regularization term, and the disentangling term is VIB. If there is no essential difference in KL divergence in $\beta$-VAE between VIB , then \alpha-TCVAE is just a modifiication of $\beta$-VAE by adding a diversity regularization term on loss function?

---

> > ### Author Response · Authors · 2023-11-23
> >
> > Thank you so much for your availability!
> >
> > A1: There is not difference between the KL diverge terms in VIB and VAE, the only difference is that in the classic VAE from Kingma and Welling, 2013, no beta term is used to enforce a bigger information bottleneck and the information theory perspective was not taken into account, while VIB rederived the same bound starting from the information bottleneck principle.
> >
> > A2: You are right, there is not difference between KL divergence in $\beta$-VAE and VIB. However, the $\alpha$-TCVAE proposes a convex combination of the KL term in $\beta$-VAE and the CEB term. In other words, it's not a simple summation, $\alpha$ must stay between 0 and 1 to get a meaningful convex bound. Moreover, it is important to note that the derivation of these terms starts from other premises (i.e., the bound is computed for TC(z,x)). Therefore, it is quite surprising that we can derive a convex combination of the known $\beta$-VAE, also known as VIB term, and the CEB term.
> >
> > We hope that the given answers, together with the new presented sections in the appendix, are convincing enough to raise the given rating.
> >
> > We are available to answer any further question, we thank the reviewer for the provided feedback and the availability.

---

### Official Review · Reviewer_wnca · 2023-10-31

**Soundness:** 2 fair
**Presentation:** 3 good
**Contribution:** 2 fair
**Rating:** 8
**Confidence:** 3

**Summary:**

The paper proposes a new method for learning disentangled representations, alpha-TCVAE.
It is based on a new lower bound for TC loss, that convexly combines VIB and CEB terms.
For alpha = 0, the alpha-TCVAE reduces to beta-VAE.
Experiments show that alpha-TCVAE brings some improvement in disentanglement quality measures and diversity of generated samples.
Comparisons with beta-TCVAE, beta-TCVAE, FactorVAE, beta-VAE+HFS, VAE and StyleGAN are provided.
An additional experiment shows usefullness of alpha-TCVAE for reinforcement learning.

**Strengths:**

1. A new approximation to the TC loss is provided.
2. In Appendix E, authors show that α-TCVAE can learn and represent generative factors that are not present in the ground-truth dataset. This is a very surprising observation, probably authors shoud consider making it more visible by moving it to the main part of the manuscript.
3. Have you used augmentations during training? Because new factors of variations (like camera elevation) mights be side effect of some augmentations like crop.
4. A comparison with the recent model β-VAE+HFS is provided.
5. The paper contains interesting insights about connections between disentanglement and diversity.
6. An application to RL is provided.
7. The paper is well written and easy to follow, the language is fine.

**Weaknesses:**

1. The improvements over other VAE-based methods are very moderate, sometimes lower than std.
2. The popular dSprites dataset is missing; also popular measures like MIG, DCI-C, DCI-I are missing.
3. The term Iθ(z, x) is not defined in Section 3.
4. The experimental setting, when a diversity is evaluated by doing traversals by +-6,8,10 std. looks strange for me.
Such points will be never sampled from the noise.
5. Where did hyperparameters from Appendix B come from? For example, Roth et al, 2023 provided a grid search for them. As far as rival methods are concerned, you can take hyperparameters from original papers (but in this case original training pipeline and architectures must be also used). If the hyperparameters for rival methods are selected arbitrary, experimental results are not valid!
6. Figure 1 is not convincing. For example, row 3 for alpha-TCVAE, contains an image with a defect: non-round violet border.
High diversity obviously can be achieved by generating wrong images.

**Questions:**

1. In Figure 2, what does green horizontal line mean?
2. In Figure 3, you write that "The scores for the images of our model, α-TCVAE, are consistently better than baseline VAE models (lower FID is better), and only slightly worse than StyleGAN."
But from Figure 3, I see that blue circles (sampling from "noise") for all VAE-based models have significantly higher FID than StyleGAN.
Probably, you compare traversal from VAE and "noise" samples from StyleGAN, but it seems not fair to me.
3. How the matrix for the Vendi score is formed?
4. FID from "traversals" image generation is much lower than from straightforward generation from noise. Do you have an explanation?
Also, it is interesting to compare VAE models via Precision/Recall (Kynkäänniemi, T., Karras, T., Laine, S., Lehtinen, J., & Aila, T. (2019). Improved precision and recall metric for assessing generative models. Advances in Neural Information Processing Systems, 32.)

**post rebuttal**. Most of my questions have been addressed, authors even did some additional computations. So, I am raising my score.

---

> ### Author Response · Authors · 2023-11-22
>
> Dear Reviewer,
>
> Thank you for your feedback, we tried to address all the points stated in the review. We hope that these answers can help the reviewer on elucidating the reviewer on some confusing parts.
> Q1: In Figure 2, what does green horizontal line mean?
> A1: The green horizontal line represents the diversity score computed using the ground truth images of the dataset. It is important to note that Table 2 in (Friedman et al, 2022) highlights how all the generative models they considered are not able to achieve a higher diversity score than the one obtained using the original dataset. By contrast, using disentangled models and for the considered datasets, this is not the case.
>
> A2: Question 2 raises a compelling point. While it's acknowledged that Style-GAN and VAEs employ distinct sampling methods, our comparison of VAE models and Style-Gan isn't intended to establish superiority of one over the other. The inclusion of a GAN-based model in our study serves a specific purpose: to demonstrate that using traversal sampling techniques can yield comparable or marginally lower visual fidelity (as measured by FID) and similar or slightly enhanced diversity (as indicated by Vendi). This finding is significant because GANs typically present greater challenges in training, including instability issues, and demand more computational resources and time. Additionally, GANs are not suitable for inferential purposes, highlighting the appeal of VAE-based models when inference capabilities are necessary.
>
> Our assessment aimed at fairness, considering the distinct training goals of GANs and VAEs. GANs are trained to sample from white noise, whereas VAEs reconstruct images from a structured latent space, sampling latents from specific points during each forward pass. Our balanced approach in sampling strategies was an effort to ensure impartiality towards both GANs and VAE-based models. Sampling with white noise inherently disadvantages VAE-based models, as it presupposes a latent space defined by a multidimensional standard Gaussian – a characteristic typical of GANs. Conversely, if one were to sample images from the VAE latent space using Style-GAN model, Style-GAN would likely exhibit poor performance as well.
>
> In summary, the reviewer's observation about the inherent unfairness in directly comparing results from these two sampling strategies is valid. However, our approach of presenting both methodologies aimed to provide an equitable evaluation of both model types.

---

> ### Author Response · Authors · 2023-11-22
>
> A3: The i,jth entry of the matrix for the Vendi score is computed as the cosine similarity between the InceptionNet feature vectors for the ith and jth images. The use of InceptionNet to extract features is specified on the top of page 6. We will add that the distance function is defined as cosine similarity.
>
> A4: This is a very interesting question. The reason why FID from 'traversals" is much lower is that using the traversals to generate images is actually the correct way of sampling images when using a VAE model. The idea is that while GAN models don't have a latent space and are trained to sample from white noise, VAE-based models have a defined latent space, which is a specific manifold with its own topology. In other words, when sampling using VAE-based models we need to be sure that the sample relies on the latent space, otherwise the image generation will not be meaningful or will be out-of-distribution (i.e., very bad FID scores).  To ensure that the samples relies on the VAE latent space, we used traversals sampling, which means that we first select a point in the latent space manifold (e.g., which basically is the encoded representation) and then we sample points close to the selected one. As a result, we make sure that the samples rely on the VAE latent space and can be used to generate meaningful images and, hence, much lower FID scores.
>
> Regarding the use of Precision and Recall as metrics, the reason why we did not use them in the first place is that, as stated in Friedman et al. 2022 -  "Compared to these approaches (i.e., Precision and Recall as metrics), the Vendi Score is a reference-free metric, measuring the intrinsic diversity of a set rather than the relationship to a reference distribution. This means that the Vendi Score should be used along side a quality metric, but can be applied in settings where there is no reference distribution." -  In other words, we were not sure of which one should be the reference distribution in our case and instead we presented Vendi Score next to FID. In these days we tried to see if we could compute them and present the results on time, but the other reviewers were interested on the sensitivity analyses of alpha, so for the sake of time we focused on that rather than on trying to understand how to compute these extra metrics. If computing them is possible, we will add these metrics on the arxiv version upon publication.  You can find the Alpha sensitivity analyses in the appendix of the updated paper.
>
> We hope that these answers, together with the new section in the appendix (Sensitivity Analyses, Relationship between CEB and Diversity,  and MIG score comparisons) can lead the reviewer to increase the grades. Thank you so much for your availability and the useful feedback.

---

> ### Author Response · Authors · 2023-11-22
> **Comments to Weaknesses**
>
> W2:
> We added MIG, DCI-C, DCI-I and DCI-D in the appendix. The presents a similar trend as the one showed by the overall DCI reported in the main paper, which is computed as geometric mean of the three DCI scores, as in Rotman et al, 2022.
>
> W3:
> Iθ(z, x) is the mutual information between z and x, we updated the main paper and added the definition.
>
> W4:
> The reason why we used only 6, 8 and 10 std is because we were running out of time and using only 3 values was faster to compute the scores. Moreover, back then we noticed that it didn't make a lot of difference in terms of results between using 1,2,4,6,8,10 and using only 6,8 and 10. We reevaluated both Vendi and FID using 1,2,4,6,8 and 10. We updated the manuscript with the new plots. As expected, the new plots do not differ that much between the one presented before.
>
> W5:
> We used the folder from Roth et al. 2023 as main codebase to run all of our experiments. For every model we used the hyperparameters used by default in their folder. Although we do not present a grid search for all the hyperparameters of all models, we believe that using the default hyperparameters from Roth codebase, which are also the one that performs the best according their gridsearch, should be a correct and fair approach.
>
> W6:
> It is true that diversity can be achieved generating wrong images, which is why we present Vendi Scores next to FID scores, that contextualize the provided Vendi Scores and makes sure that high values of Diversity are not obtained because the model generates wrong images.

---

### Meta-Review · Area_Chair_RCW6 · 2023-12-17

**Metareview:**

The paper a new total correlation (TC) lower bound specified as a convex combination of the known VIB and CEB terms. The intuition is maximize disentanglement and informativeness in the latent variables. In empirical studies, the author finds their formulation learns better disentangled representations compared to the original TCVAE and other baselines. The reviewers appreciated the exposition of the paper, and some of the empirical results (such as application to RL). There were also concerns regarding the incremental nature  and distinction with existing the approaches (esp. relative to beta-TCVAE), as well as some of the evaluation protocols related to metrics and dataset coverage. The authors addressed most of these concerns.

**Justification For Why Not Higher Score:**

2 out of 3 reviewers still believe the overall contribution is incremental over existing works.

**Justification For Why Not Lower Score:**

General concerns of reviewers all addressed. Good insights and decent empirical results.

---

### Decision · Program_Chairs · 2024-01-16

Accept (poster)